# Budget-Feasible Mechanisms for Submodular Welfare Maximization in Procurement Auctions

**Shuang Cui** [1]   **He Huang** [1]   **Yu-e Sun** [2]   **Chen Xue** [1]

## Abstract

Budget-feasible procurement auctions play a pivotal role in various AI-driven marketplaces, such as data acquisition and crowdsourcing, where a buyer with a limited budget seeks to procure services from strategic sellers with private costs. While numerous budget-feasible mechanisms have been proposed for the classic objective of maximizing the buyer's valuation, the more challenging and economically significant objective of social welfare maximization has only recently been studied, and existing approaches still sacrifice budget feasibility, thereby limiting their practical applicability. In this paper, we bridge this gap by proposing BFM-SWM, the first budget-feasible mechanism with provable approximation guarantees for submodular welfare maximization in procurement auctions. Our mechanism satisfies standard economic properties, including truthfulness, individual rationality, and non-negative auctioneer surplus. As a by-product, we develop BFM-VM, a variant tailored for valuation maximization, which achieves a deterministic approximation ratio of $1/(12 + 4\sqrt{3})$ for general submodular functions, substantially improving upon the best-known deterministic ratio of $1/64$ established by [Balkanski et al., SODA 2022], while reducing the running time from $\mathcal{O}(n^2 \log n)$ to $\mathcal{O}(n \log n)$. Extensive experiments demonstrate the efficiency and effectiveness of our mechanisms.

## 1. Introduction

Procurement auctions, executed by governments or private companies to purchase goods and services from suppliers, have been widely adopted across various domains, including crowdsourcing markets (Singer & Mittal, 2013; Han et al., 2025), influence maximization (Kempe et al., 2003), industrial procurement (Bichler et al., 2005), and data acquisition (Fallah et al., 2024). In practice, the available budget is typically limited, imposing a cap on the total payment to suppliers (Leonardi et al., 2021).

(Singer, 2010) proposed a classic procurement auction setting involving a single auctioneer and a set $\mathcal{N}$ of $n$ sellers, each owning a single item. The auctioneer is endowed with a submodular valuation function $v : 2^{\mathcal{N}} \to \mathbb{R}_{\geq 0}$, where $v(S)$ represents the value the auctioneer derives from acquiring items from sellers $S \subseteq \mathcal{N}$. Each seller $u \in \mathcal{N}$ has a private cost $c(u) \geq 0$, representing the minimum payment required for providing the item. Furthermore, (Singer, 2010) pioneered the design of *budget-feasible mechanisms* under this setting, aiming to approximately achieve **valuation maximization** (i.e., $\max_{S \subseteq \mathcal{N}} v(S)$) while satisfying truthfulness, individual rationality, computational efficiency, and budget feasibility. Following this, a rich line of work has focused on designing more effective budget-feasible mechanisms to improve the approximation ratio $\gamma_v := v(S)/v(O)$, where $S$ denotes the solution returned by the mechanism and $O$ denotes the optimal solution. For general (not necessarily monotone) submodular valuation functions, the current best randomized approximation ratio is $0.0856$, proposed by (Han et al., 2025), while the best deterministic ratio is $0.0156$, proposed by (Balkanski et al., 2022).

Recently, (Deng et al., 2025) considered a different objective in procurement auctions, namely **welfare maximization** (i.e., $\max_{S \subseteq \mathcal{N}} \big( v(S) - c(S) \big)$). This objective can be viewed as analogous to the notion of gains-from-trade in the bilateral trade literature, as it measures the net social value generated by running the procurement auction. Unlike valuation maximization, which may prioritize high-value but exorbitantly costly items, welfare maximization inherently promotes cost-efficiency and allocative efficiency. It tends to favor sellers who provide high marginal value relative to their costs, which is critical for sustainable procurement.

From a technical perspective, welfare maximization is substantially more challenging than valuation maximization, since its objective $v(\cdot) - c(\cdot)$ may be negative and cannot

[1] School of Computer Science and Technology, Soochow University, Suzhou 215006, China  [2] School of Rail Transportation, Soochow University, Suzhou 215006, China . Correspondence to: He Huang <huangh@suda.edu.cn>.

*Proceedings of the 43rd International Conference on Machine Learning*, Seoul, South Korea. PMLR 306, 2026. Copyright 2026 by the author(s).

be directly evaluated as $c(\cdot)$ is private. As a result, existing budget-feasible mechanisms, which typically rely on non-negativity and direct observability of the whole objective function, no longer apply. Notably, (Nikolakaki et al., 2021) pointed out that even in the simplified setting where costs are public, it is impossible to obtain a constant-factor multiplicative approximation in polynomial time for welfare maximization. Consequently, prior work (Nikolakaki et al., 2021; Sviridenko et al., 2017; Harshaw et al., 2019; Cui et al., 2023; Feldman, 2021) typically adopts a weaker approximation notion, seeking a solution $S$ satisfying $v(S) - c(S) \geq \gamma_w \cdot v(O) - c(O)$, where $\gamma_w$ denotes the approximation ratio and $O$ denotes the optimal solution.

Using Myerson's lemma (Myerson, 1981) and the form of sealed-bid auction, (Deng et al., 2025) designs mechanisms tailored for welfare maximization that satisfy truthfulness, individual rationality, computational efficiency, and non-negative auctioneer surplus. However, they assume there is no budget constraint and thus significantly simplify the mechanism design task[1], which also renders their mechanisms impractical for real-world scenarios where the buyer's purchasing power is inherently finite. Therefore, designing budget-feasible mechanisms for submodular welfare maximization in procurement auctions remains a significant and challenging open problem.

## 1.1. Our Contributions

In this paper, we address the above open problem by proposing novel mechanisms for welfare maximization. Our main contributions can be summarized as follows:

- We propose BFM-SWM, a budget-feasible mechanism for welfare maximization in procurement auctions that satisfies truthfulness, individual rationality, and non-negative auctioneer surplus. Under the commonly adopted approximation notion defined above, BFM-SWM achieves a 0.0328-approximation for general (not necessarily monotone) submodular functions and a 0.0877-approximation for monotone submodular functions, up to an arbitrarily small additive error.

  To the best of our knowledge, our BFM-SWM is the first budget-feasible mechanism with provable approximation guarantees for welfare maximization in procurement auctions.

- As a by-product, we develop a variant of the mech-

anism, named BFM-VM, tailored for the valuation maximization objective while preserving budget feasibility, truthfulness, and individual rationality. Notably, BFM-VM achieves a deterministic approximation ratio of $1/(12 + 4\sqrt{3}) \approx 0.0528$ for general submodular functions, which significantly improves upon the best-known deterministic ratio of $1/64 \approx 0.0156$ proposed by (Balkanski et al., 2022) in SODA 2022, while reducing the running time from $\mathcal{O}(n^2 \log n)$ to $\mathcal{O}(n \log n)$.

- We conduct extensive experiments to evaluate the empirical performance of our mechanisms. The results convincingly validate their effectiveness and efficiency.

## 1.2. Challenges and Techniques

As mentioned above, welfare maximization is substantially more challenging than valuation maximization. The key obstacles are that the welfare objective $v(\cdot) - c(\cdot)$ may be negative and cannot be directly evaluated as $c(\cdot)$ is private. As a result, existing budget-feasible mechanisms, which typically rely on non-negativity and direct observability of the whole objective function, no longer apply. We note that only (Deng et al., 2025) has successfully designed a mechanism with provable approximation guarantees for welfare maximization, albeit by sacrificing budget-feasibility. It is worth noting that, as indicated by (Singer, 2010; Chen et al., 2011; Liu et al., 2024), designing budget-feasible mechanisms is significantly more challenging than their unconstrained counterparts. The budget constraint not only restricts the feasible solution space but also couples the allocation and payment rules, preventing the straightforward allocation-then-payment construction typically employed in unconstrained settings such as (Deng et al., 2025).

We propose a novel budget-feasible mechanism for welfare maximization, named BFM-SWM, which overcomes the above challenges while satisfying the desired economic properties and achieving provable approximation guarantees. Our BFM-SWM employs a geometrically increasing threshold that is initialized to an arbitrarily small value independent of the true welfare values, and iteratively increases until no seller can satisfy the threshold requirement. This progressive threshold serves a dual purpose: acting as a key factor in determining price offers and as a benchmark for evaluating the welfare of candidate solutions. This design effectively circumvents the need to directly evaluate the exact welfare of candidate sets, while still theoretically guaranteeing that the generated sets possess sufficiently high welfare. Moreover, since private costs hinder the bounding of individual objective values, we introduce a singleton candidate solution to capture the critical single elements. The seller added to this candidate receives temporal protection against descending price offers, preventing premature exits and preserving procurement opportunities for their high-

---

[1]As indicated by (Singer, 2010; Chen et al., 2011; Liu et al., 2024), designing budget-feasible mechanisms is significantly more challenging than their unconstrained counterparts. The budget constraint not only restricts the feasible solution space but also couples the allocation and payment rules, preventing a straightforward allocation-then-payment construction that is typically used in unconstrained settings.

value item. Additionally, we introduce a control parameter $\beta$ within the pricing rule to regulate the value-to-payment ratio. This ensures that the valuation of the selected outcome is at least a $\beta$-multiple of its payment, effectively enforcing the non-negativity of the resulting welfare and avoiding buyer deficit. Finally, distinct from the sealed-bid format adopted in (Deng et al., 2025), BFM-SWM takes the form of descending clock auctions, thereby offering stronger strategyproofness properties.

The generality and effectiveness of our approach extend beyond welfare maximization, as evidenced by a by-product deterministic variant, BFM-VM, for the valuation maximization problem. In particular, BFM-VM achieves a deterministic approximation ratio of $1/(12+4\sqrt{3})$, which substantially improves upon the best-known deterministic ratio of $1/64$ by (Balkanski et al., 2022) in SODA 2022, while simultaneously reducing the running time from $\mathcal{O}(n^2 \log n)$ to $\mathcal{O}(n \log n)$. This improvement stems from a design and theoretical analysis fundamentally different from that of (Balkanski et al., 2022). Specifically, BFM-VM constructs disjoint candidate solutions only via threshold-based filtering, rather than relying on additional computationally heavier subroutines such as greedy selection based on marginal gains and unconstrained submodular maximization algorithms. In summary, our techniques are not only robust enough to handle the more challenging welfare maximization, but also lead to stronger guarantees for the classical valuation maximization.

## 2. Related Work

**Regularized Submodular Maximization.** The welfare maximization problem, where the objective takes the form of a submodular valuation function minus cost (i.e., $v(\cdot) - c(\cdot)$), has been extensively studied in the algorithmic literature under the assumption of public costs, a domain known as regularized submodular maximization. Since obtaining a constant-factor multiplicative approximation for this problem in polynomial time is theoretically impossible, researchers (Sviridenko et al., 2017; Cui et al., 2023; Feldman, 2021) typically adopt a weaker approximation notion, aiming to find a solution $S$ satisfying $v(S) - c(S) \geq \gamma_w \cdot v(O) - c(O)$, where $\gamma_w$ denotes the approximation ratio and $O$ denotes the optimal solution. Notable algorithms in this category include Distorted Greedy (Harshaw et al., 2019), ROI Greedy (Jin et al., 2021), and Cost-Scaled Greedy (Nikolakaki et al., 2021). However, these algorithms are incompatible with procurement auctions where costs are private, due to their reliance on public cost information and lack of economic properties.

To bridge this gap, recent work (Deng et al., 2025) proposed a framework for converting regularized submodular maximization algorithms into mechanisms with economic prop-

erties. Nevertheless, their approach necessitates sacrificing budget feasibility, thereby limiting its practical applicability.

**Budget-Feasible Mechanisms for Valuation Maximization.** Valuation maximization can be regarded as a special, more tractable case of welfare maximization, where the objective is restricted to the valuation term $v(\cdot)$. Early mechanisms for this problem relied on sealed-bid formats to ensure truthfulness (Singer, 2010; Chen et al., 2011; Leonardi et al., 2021; Amanatidis et al., 2017; Jalaly & Tardos, 2021; Amanatidis et al., 2019; Neogi et al., 2024); however, these designs are often criticized for their limited transparency and complex strategic implications. Subsequently, (Milgrom & Segal, 2020) introduced the descending clock auction format that offers obvious strategyproofness, a stronger and more intuitive notion of truthfulness. Following this, a series of studies have adopted the clock auction format to design budget-feasible mechanisms for submodular valuation maximization (Balkanski et al., 2022; Han et al., 2023; Huang et al., 2023; Han et al., 2025). However, as discussed in Section 1.2, these existing mechanisms are unsuitable for welfare maximization, because they typically rely on the non-negativity and direct observability of the whole objective function, properties that do not hold for the welfare objective due to private costs. To the best of our knowledge, no prior work has developed budget-feasible mechanisms with provable approximation guarantees for submodular welfare maximization in procurement auctions, regardless of auction format (sealed-bid or descending clock).

## 3. Preliminaries

We consider a procurement auction setting with a single auctioneer and a set $\mathcal{N}$ of $n$ sellers, each owning a single item. The auctioneer is endowed with a valuation function $v : 2^{\mathcal{N}} \to \mathbb{R}_{\geq 0}$, where $v(S)$ represents the value the auctioneer derives from acquiring the items offered by the sellers $S \subseteq \mathcal{N}$. Each seller $u \in \mathcal{N}$ has a private cost $c(u) \geq 0$, representing the minimum payment required for providing its item to the auctioneer. We consider the valuation function $v(\cdot)$ to be normalized (i.e., $v(\emptyset) = 0$), submodular, and not necessarily monotone. A function $v(\cdot)$ is submodular if it satisfies the diminishing returns property: for all subsets $X \subseteq Y \subseteq \mathcal{N}$ and any element $u \in \mathcal{N} \setminus Y$, it holds that $v(u \mid Y) \leq v(u \mid X)$, where $v(S \mid T) \triangleq v(S \cup T) - v(T)$ denotes the marginal contribution of a set $S$ with respect to $T$. For notational simplicity, we define $v(u) \triangleq v(\{u\})$ for any element $u \in \mathcal{N}$. For any subset $X \subseteq \mathcal{N}$, we let $c(X) \triangleq \sum_{u \in X} c(u)$ and $p(X) \triangleq \sum_{u \in X} p(u)$ denote the total cost and total payment, respectively. Additionally, for any integer $i \geq 1$, we denote the set $\{1, \ldots, i\}$ by $[i]$.

Our objective is to design a computationally efficient mechanism that determines a winning seller set $S \subseteq \mathcal{N}$ and corresponding payment $p(S)$ to maximize the social wel-

fare, defined as $v(S) - c(S)$. The mechanism must satisfy the following desired economic properties: (1) *Truthfulness*; (2) *Individual Rationality*; (3) *Non-negative Auctioneer Surplus*, i.e., $v(S) - p(S) \geq 0$; and (4) *Budget Feasibility*, i.e., $p(S) \leq B$ for a given budget $B$. Note that the welfare objective $v(\cdot) - c(\cdot)$ preserves submodularity but may take negative values. Crucially, existing literature establishes that no polynomial-time algorithm can achieve a multiplicative approximation guarantee for maximizing a potentially negative submodular function, regardless of constraints (Nikolakaki et al., 2021). Consequently, following prior work (Nikolakaki et al., 2021; Sviridenko et al., 2017; Harshaw et al., 2019; Cui et al., 2023; Feldman, 2021), we adopt a weaker approximation notion and aim to find a solution $S$ satisfying $v(S) - c(S) \geq \gamma_w \cdot v(O) - c(O)$ where $\gamma_w$ denotes the approximation ratio and $O$ denotes the optimal solution.

# 4. Budget-Feasible Mechanisms for Welfare Maximization

In this section, we present our budget-feasible mechanism for submodular welfare maximization in procurement auctions. Owing to its general framework, the mechanism can be adapted to valuation maximization via several modifications, which is presented in Section 5.

To adhere to space limitations and ensure a clear narrative flow, we defer the complete proofs of most lemmas and theorems to Appendix A, presenting only the key proof ideas in the main text.

### 4.1. Mechanism Design

As outlined in Algorithm 1, our mechanism BFM-SWM operates as a multi-round descending clock auction that maintains multiple candidate seller sets to maximize social welfare. The mechanism first iterates through the entire set of sellers $\mathcal{N}$, offering each seller the full budget $B$. Sellers who accept this initial offer constitute the active seller set $R$ for the subsequent auction process (Line 1). Then, the mechanism initializes the threshold $\rho_0$ as a small value $\epsilon/\alpha$ (where $\epsilon$ is an input error parameter), which will serve as a key factor in determining price offers in subsequent auction rounds. Additionally, the singleton candidate solution $u^*$, which captures critical individual sellers, is initialized to the empty set. The mechanism also initializes an owner index for each active seller, which is used to keep the different aggregate candidate sequences disjoint (Line 2).

Next, the mechanism enters a multi-round iterative auction phase that continues until all active sellers in $R$ have been either tentatively selected in the recent two rounds or have exited the auction (Lines 3-25). At the beginning of each round $t$, the threshold $\rho_t$ is updated multiplicatively by a factor of $\alpha > 1$. This geometric increase is designed to

---

**Algorithm 1** A Budget-Feasible Mechanism for Submodular Social Welfare Maximization (BFM-SWM)

---

**input** budget $B$, parameters $\alpha > 1, \beta > 1, \epsilon > 0$, and number of candidate set sequences $\ell \in \{1, 2\}$

1: Let $R \leftarrow \{u \in \mathcal{N} \mid u \text{ accepts price } B\}$ and $p(u) \leftarrow B$ for each $u \in R$

2: Initialize $t \leftarrow 0$, $\rho_t \leftarrow \epsilon/\alpha$, $u^* \leftarrow \emptyset$, and $\text{owner}(u) \leftarrow 0$ for each $u \in R$

3: **repeat**

4:   $t \leftarrow t + 1$; $\rho_t \leftarrow \alpha \cdot \rho_{t-1}$; $S_{i,t} \leftarrow \emptyset$ for each $i \in [\ell]$

5:   **for** $u \in R \setminus \left( \bigcup_{i=1}^{\ell} S_{i,t-1} \cup u^* \right)$ **do**

6:     **if** $\text{owner}(u) \neq 0$ **then**

7:       $j \leftarrow \text{owner}(u)$

8:     **else**

9:       $j \leftarrow \arg\max_{i \in [\ell]} v(u \mid S_{i,t})$

10:    **end if**

11:    Update $p(u) \leftarrow \min \left\{ p(u), \frac{v(u|S_{j,t})}{\beta + \rho_t/B} \right\}$

12:    **if** $u$ accepts price $p(u)$ **then**

13:      **if** $v(S_{j,t} \cup \{u\}) - p(S_{j,t} \cup \{u\}) > \rho_t$ **then**

14:        $u^* \leftarrow \{u\}$; **break**

15:      **else**

16:        $S_{j,t} \leftarrow S_{j,t} \cup \{u\}$;

17:        **if** $\text{owner}(u) = 0$ **then**

18:          $\text{owner}(u) \leftarrow j$

19:        **end if**

20:      **end if**

21:    **else**

22:      $R \leftarrow R \setminus \{u\}$;

23:    **end if**

24:   **end for**

25: **until** $R \setminus \left( \bigcup_{i=1}^{\ell} (S_{i,t-1} \cup S_{i,t}) \cup u^* \right) = \emptyset$

26: $M \leftarrow t$; $S^* \leftarrow \arg\max_{A \in \{S_{i,t} \mid i \in [\ell], t \in \{M-1, M\}\} \cup \{u^*\}} \{v(A) - p(A)\}$

**output** $S^*$ and $p(u)$ for each $u \in S^*$

---

progressively lower the prices offered to sellers, aiming to reduce payments closer to the sellers' private costs and thereby enhance social welfare. The specific value of $\alpha$ is determined in our theoretical analysis to optimize the approximation ratio. After initializing the current round's candidate sets $\{S_{i,t}\}_{i=1}^{\ell}$ to be empty, we proceed to iterate through the active sellers in $R$, excluding those who were already selected in the previous round $t - 1$ (Lines 5-24).

For each seller $u$, the mechanism first determines which candidate sequence should process $u$. To ensure that different aggregate candidate sequences are disjoint, the mechanism maintains an owner index for each active seller. Initially, every seller is unassigned. If $u$ has already been assigned to a sequence in a previous round, then $u$ is routed to its owner sequence. Otherwise, the mechanism greedily identifies the candidate set $S_{j,t}$ to which $u$ provides the largest marginal

gain (Line 9). If an unassigned seller is accepted and added to $S_{j,t}$, its owner is permanently set to $j$ (Line 16). Based on the marginal gain $v(u \mid S_{j,t})$, the budget $B$, the current threshold $\rho_t$, and a value-to-payment ratio parameter $\beta$ (which ensures the value obtained by the auctioneer is at least $\beta$ times the payment and whose value is determined in our theoretical analysis to optimize the approximation ratio), the mechanism computes a new lower price (Line 11). If a seller $u$ rejects the new offer, they permanently exit the auction (since prices only decrease) and are removed from the active set $R$ (Line 22). If $u$ accepts, the mechanism checks a break condition: whether adding $u$ to $S_{j,t}$ would cause the auctioneer's surplus for that set to exceed the current threshold $\rho_t$ (Line 13). If this condition is triggered, we store the element to $u^*$. The seller added to this singleton candidate receives temporal protection against descending price offers, preventing premature exit and preserving the procurement opportunity for the high-value item. The inner loop then terminates immediately for the current round. This break condition is crucial for ensuring budget feasibility, as detailed in the proof of Theorem 4.1. If this condition is not triggered, $u$ is added to the candidate set $S_{j,t}$ (Line 16).

Upon termination of the outer loop, the mechanism selects the seller set $S^*$ that maximizes welfare among all candidate sets generated in the last two rounds (including the singleton candidate solution $u^*$). It outputs $S^*$ as the winning set and assigns each seller $u \in S^*$ the most recently posted price $p(u)$.

### 4.2. Theoretical Analysis

In this section, we analyze the performance guarantees of the mechanism for general (not necessarily monotone) submodular valuation functions by setting the number of candidate set sequences to $\ell = 2$. We derive an improved approximation ratio for the special case of monotone submodular valuation functions with $\ell = 1$ in the following Section 4.3.

Since any seller whose cost exceeds the budget $B$ cannot belong to any feasible solution, such sellers are removed by Line 1 of our mechanism. Hence, we do not consider these sellers in the subsequent analysis. Consider any candidate set $S_{i,t}$ during a round $t \in [M]$. For any seller $u$ processed in round $t$, we denote by $S_{i,t}^u$ the state of $S_{i,t}$ immediately before $u$ is processed in Line 11, for every $i \in [\ell]$. If $u$ is not processed in round $t$, we define $S_{i,t}^u = S_{i,t}$, namely, the final state of the set at the end of round $t$.

We first show that the proposed mechanism satisfies the standard economic properties and is computationally efficient.

**Theorem 4.1.** *BFM-SWM is a budget-feasible mechanism that satisfies obvious strategyproofness (which implies truthfulness), individual rationality, and the non-negative auctioneer surplus condition. Moreover, it runs in* $\mathcal{O}(n \log \frac{OPT}{\epsilon})$ *time, where OPT denotes the welfare of the*

*optimal solution.*

We next analyze the approximation ratio of the mechanism. To this end, we need to upper bound the welfare of the optimal solution $O$ using the solution output by our mechanism. The main challenge lies in accounting for the welfare loss incurred by the elements in $O$ that are not included in any of our candidate sets $\{S_{i,t}\}_{t=1}^{M}$. We address this by first partitioning such elements according to the reasons for their exclusion, as detailed below.

**Definition 4.2.** Let $U_i = \bigcup_{t=1}^{M} S_{i,t}$ for $i \in \{1,2\}$. We partition the set of elements in the optimal solution $O$ that are not included in $U_1$, i.e., $O \setminus U_1$, into the following three categories.

1. **Elements first added to the other aggregate candidate sequence $U_2$.** For each round $t$, let $O_{2,t}^{>}$ denote the subset of elements in $O \setminus U_1$ whose first successful insertion into $U_2$ occurs in round $t$.

2. **Elements rejected before any successful insertion.** These elements have not been successfully inserted into either aggregate candidate sequence before they reject an offered price. For each round $t$, let $O_{1,t}^{<}$ and $O_{2,t}^{<}$ denote the subsets of elements in $O \setminus (U_1 \cup U_2)$ that reject the offered price in round $t$ when they are considered for $S_{1,t}$ and $S_{2,t}$, respectively.

3. **The final singleton element.** Let $O^* = (u^* \cap O) \setminus (U_1 \cup U_2)$ denote the final singleton element, if it belongs to $O$ and has not been successfully inserted into either aggregate candidate sequence.

Based on the above, we obtain the complete partition

$$O \setminus U_1 = \bigcup_{t=1}^{M} \left( O_{1,t}^{<} \cup O_{2,t}^{<} \cup O_{2,t}^{>} \right) \cup O^*.$$

By symmetry, $O \setminus U_2$ can be partitioned as

$$O \setminus U_2 = \bigcup_{t=1}^{M} \left( O_{2,t}^{<} \cup O_{1,t}^{<} \cup O_{1,t}^{>} \right) \cup O^*.$$

Using this partition, we can derive an upper bound on a scaled welfare expression of the optimal solution, as shown in Lemma 4.3. The proof relies on two key insights: (1) Elements in $O_{i,t}^{<}$ ($i \in [\ell]$) rejected the price offers determined by their marginal gain, the budget, the threshold, and the value-to-payment ratio parameter, which implies that their "cost-efficiency" (i.e., the ratio of marginal welfare to cost) is insufficient relative to the threshold. Thus, the total welfare loss caused by excluding these elements can be bounded by the final threshold $\rho_M$. (2) Elements in $O_{i,t}^{>}$ ($i \in [\ell]$) were not selected for a specific candidate set

because, under the greedy strategy (Line 9), they contributed more significant marginal value to a "sibling" candidate set. Therefore, the welfare loss caused by excluding these elements can be directly charged to the welfare accumulated by the corresponding sibling candidate sets.

**Lemma 4.3.** We have $v(O) - 2\beta \cdot c(O) \leq 2\rho_M + 2\sum_{i=1}^{\ell}\sum_{t=1}^{M} v(S_{i,t}) + 2v(u^*)$.

To establish the approximation ratio, it now suffices to bound the right-hand side of the inequality in the above lemma. We first address the trivial case where the mechanism terminates within one round, i.e., $M \leq 1$. In this case, we can directly derive a simple upper bound, as stated in Lemma 4.4.

**Lemma 4.4.** If $M \leq 1$, then $v(O) - 2\beta \cdot c(O) \leq 2\epsilon + 2\sum_{i=1}^{\ell} v(S_{i,1}) + 2v(u^*)$.

We now turn to the non-trivial case where $M \geq 2$. Our analysis begins by deriving an upper bound on the final threshold $\rho_M$, as presented in Lemma 4.5. This result fundamentally relies on the break condition enforced in Line 13 of the mechanism.

**Lemma 4.5.** If $M \geq 2$, then $\rho_M \leq 2\alpha \left(v(S^*) - p(S^*)\right)$.

*Proof.* Consider round $M - 1$. At least one of the candidate sets $\{S_{i,M-1}\}_{i=1}^{\ell}$ must have triggered the break instruction upon the evaluation of some element; otherwise, we would have $R \setminus \left(\bigcup_{i=1}^{\ell}(S_{i,M-2} \cup S_{i,M-1}) \cup u^*\right) = \emptyset$, which would imply that round $M$ is never reached, contradicting the definition of $M$. Therefore, $u^*$ must be non-empty. Let $S'$ denote the candidate set that triggered the break condition with the final element in $u^*$. Regardless of whether this occurred in round $M - 1$ or $M$, the break condition implies $\rho_M/\alpha = \rho_{M-1} \leq v(S' \cup \{u^*\}) - p(S' \cup \{u^*\}) \leq v(S') - p(S') + v(u^*) - p(u^*)$ due to submodularity. Using $S^* \leftarrow \arg\max_{A \in \{S_{i,t}|i\in[\ell],t\in\{M-1,M\}\}\cup\{u^*\}}\{v(A) - p(A)\}$ completes the proof.

$\square$

We next proceed to bound the remaining terms on the right-hand side of the inequality in Lemma 4.3, as established in Lemma 4.7. The proof leverages three key design features of our mechanism: (1) the value-to-payment ratio parameter $\beta$ ensures that the valuation of any candidate solution is at least $\beta$ times its payment, which allows us to bound its valuation by its welfare (as shown by Lemma 4.6); (2) due to the break condition in Line 13, the welfare of each candidate solution in a given round is upper bounded by the corresponding threshold; and (3) the thresholds grow geometrically by a factor of $\alpha$, which implies that the sum of thresholds across all rounds is dominated by the final term $\rho_M$, which we have already bounded in Lemma 4.5.

**Lemma 4.6.** For any candidate solution $S_{i,t}$ ($i \in [\ell], t \in [M]$) and the singleton set $u^*$, we have $v(A) \leq \frac{v(A)-p(A)}{1-1/\beta} \leq \frac{v(A)-c(A)}{1-1/\beta}$ for $A \in \{S_{i,t}\} \cup \{u^*\}$.

*Proof.* Consider any element $u$ that accepts the price offered by $S_{i,t}$ ($t \in [M], i \in [\ell]$). By the pricing rule in Line 11, we have $c(u) \leq p(u) \leq \frac{v(u|S_{i,t}^u)}{\beta+\rho_t/B}$. Rearranging this inequality yields $v(u \mid S_{i,t}^u) - \beta \cdot p(u) \geq \frac{\rho_t}{B} \cdot p(u) \geq 0$. Summing over all elements in $S_{i,t}$ yields $v(S_{i,t}) - \beta \cdot p(S_{i,t}) \geq 0$, and hence $p(S_{i,t}) \geq \beta \cdot p(S_{i,t}) \geq \beta \cdot c(S_{i,t})$. This yields $v(S_{i,t}) \leq \frac{v(S_{i,t})-p(S_{i,t})}{1-1/\beta} \leq \frac{v(S_{i,t})-c(S_{i,t})}{1-1/\beta}$. We can use similar reasoning to get $v(u^*) \leq \frac{v(u^*)-p(u^*)}{1-1/\beta} \leq \frac{v(u^*)-c(u^*)}{1-1/\beta}$ as $v(u^*) \geq v(u^* \mid S_{i,t}^{u^*})$. $\square$

**Lemma 4.7.** It holds that $\sum_{i=1}^{\ell}\sum_{t=1}^{M} v(S_{i,t}) \leq \frac{2\ell \cdot \alpha(v(S^*)-p(S^*))}{(\alpha-1)(1-1/\beta)}$.

Combining Lemmas 4.3–4.7, we are now ready to establish the approximation guarantee of our mechanism BFM-SWM.

**Theorem 4.8.** By setting $\ell = 2$, $\alpha = 1 + \frac{2\sqrt{6}}{3}$, and $\beta = 4$, BFM-SWM outputs a winning set $S^*$ satisfying $v(S^*) - c(S^*) \geq 0.0328 \cdot v(O) - c(O) - \epsilon/4$ for the general (not necessarily monotone) submodular valuation function $v(\cdot)$, where $\epsilon > 0$.

## 4.3. Improved Approximation Guarantees for Monotone Valuation Functions

In this section, we consider the special case where the valuation function $v(\cdot)$ is monotone. While the theoretical analysis presented in the previous section remains valid for this case, the monotonicity of the valuation function allows us to simplify the mechanism and achieve stronger approximation guarantees.

Specifically, under monotonicity, the mechanism only needs to maintain a single sequence of candidate sets, i.e., $\ell = 1$. Consequently, the sets corresponding to the second candidate sequence $O_{2,t}^<$ and $O_{2,t}^>$ (from Definition 4.2) are empty for all $t$. Furthermore, monotonicity implies $v\left(\bigcup_{t=1}^{M} S_{1,t} \cup O\right) \geq v(O)$. These observations lead to a tighter bound compared to Lemma 4.3, as established in the following lemma.

**Lemma 4.9.** It holds that $v(O) - \beta \cdot c(O) \leq \rho_M + v\left(\bigcup_{t=1}^{M} S_{1,t}\right) + v(u^*)$.

Combining Lemma 4.9 with the remaining lemmas that continue to hold in the monotone setting (Lemmas 4.5–4.7), we derive an improved approximation ratio for monotone submodular valuation functions, as stated in Theorem 4.10.

**Theorem 4.10.** By setting $\ell = 1$, $\alpha = 1 + \frac{\sqrt{6}}{2}$, and $\beta = 3$, BFM-SWM outputs a winning set $S^*$ satisfying $v(S^*) -$

$c(S^*) \geq 0.0877 \cdot v(O) - c(O) - \epsilon/3$ *for the monotone submodular valuation function* $v(\cdot)$, *where* $\epsilon > 0$.

## 5. Variant for Valuation Maximization

From a problem definition perspective, the *Valuation Maximization* in procurement auctions (i.e., $\max_{S \subseteq \mathcal{N}} v(S)$) can be viewed as a special case of *Welfare Maximization* (i.e., $\max_{S \subseteq \mathcal{N}} (v(S) - c(S))$) when the cost function $c(\cdot) \equiv 0$. However, while the cost is absent from the valuation maximization goal, the actual costs of sellers are non-zero and private, which continue to dictate feasibility and pricing. Thus, the theoretical guarantees derived in the previous sections for our BFM-SWM do not immediately apply to this problem.

Nevertheless, since valuation maximization is easier to handle than welfare maximization as we discussed in Section 1, BFM-SWM can be adapted to valuation maximization via several simple yet effective modifications, while preserving its feasibility and key economic properties. In this section, we formally introduce the variant denoted as **BFM-VM**, and show that it can achieve a deterministic approximation ratio of $1/(12 + 4\sqrt{3})$, which significantly improves upon the current best-known deterministic ratio of $1/64$ proposed by (Balkanski et al., 2022). The BFM-VM variant is obtained by introducing the following specific modifications to BFM-SWM (the full pseudocode is provided in Algorithm 2 of Appendix B):

1. Replace the initialization in Line 2 of Algorithm 1 with "$t \leftarrow 1; \rho_t \leftarrow \max_{u \in R} v(u); S_{i,t} \leftarrow \emptyset$ for each $i \in [\ell]$; let $a \in \arg\max_{u \in R} v(u); S_{1,t} \leftarrow \{a\}; \text{owner}(u) \leftarrow 0$ for each $u \in R$ and $\text{owner}(a) \leftarrow 1$".

2. The singleton candidate set $u^*$ is omitted.

3. The value-to-payment ratio parameter $\beta$ is set to 0.

4. The break condition in Line 13 is replaced with $v(S_{j,t} \cup \{u\}) > \rho_t$.

These modifications are primarily motivated by the following considerations: (1) they refocus the optimization objective on valuation rather than social welfare; (2) the valuation objective is easier to evaluate than the welfare objective (the cost terms in welfare functions are private and thus cannot be directly queried via an oracle), which enables us to employ more effective seller set filtering based on the threshold.

The economic properties established in Theorem 4.1 continue to hold for this variant except for the non-negative auctioneer surplus property. We next analyze its approximation guarantee. Using arguments similar to those in Lemmas 4.3–4.7, we can derive analogous bounds of BFM-VM for valuation maximization, as formalized in Lemmas 5.1–5.3.

**Lemma 5.1.** $v(O) \leq 2\rho_M + 2\sum_{i=1}^{\ell}\sum_{t=1}^{M} v(S_{i,t})$.

**Lemma 5.2.** *We have* $v(S^*) + \rho_1 \geq \rho_{M-1}$, *which can yield* $\frac{\alpha^2}{\alpha-1} v(S^*) \geq \rho_M$.

**Lemma 5.3.** $\sum_{i=1}^{\ell}\sum_{t=1}^{M} v(S_{i,t}) \leq \frac{4\alpha-2}{\alpha-1} v(S^*)$

Combining all the above, we can directly obtain the approximation ratio of BFM-VM for valuation maximization:

**Theorem 5.4.** *By setting* $\ell = 2$ *and* $\alpha = 1 + \sqrt{3}$, *the deterministic mechanism BFM-VM outputs a winning set* $S^*$ *satisfying* $v(S^*) \geq v(O)/(12 + 4\sqrt{3})$ *for the general (not necessarily monotone) submodular valuation function* $v(\cdot)$. *Moreover, it runs in* $\mathcal{O}(n \log n)$ *time.*

## 6. Experiments

In this section, we empirically evaluate the effectiveness and efficiency of our BFM-SWM mechanism. We conduct extensive experiments on a real-world application for submodular welfare maximization in procurement auctions (i.e., influence maximization). Additionally, we compare the empirical performance of our by-product variant BFM-VM against state-of-the-art mechanisms for valuation maximization in a crowdsourcing application, as shown in Appendix C. The experimental results strongly confirm the superiority of our framework for both welfare and valuation maximization objectives.

### 6.1. Baselines and Implementation

To the best of our knowledge, (Deng et al., 2025) is the only existing work that has successfully developed mechanisms with provable approximation guarantees for welfare maximization in procurement auctions, albeit without considering budget feasibility. They proposed a framework for converting regularized submodular maximization algorithms into mechanisms with economic properties including truthfulness, individual rationality and non-negative auctioneer surplus. We adopt the algorithms with provable approximation guarantees listed in their work as our baselines, including Distorted Greedy (Harshaw et al., 2019), ROI Greedy (Jin et al., 2021), and Cost-Scaled Greedy (Nikolakaki et al., 2021). These algorithms are converted into mechanisms using the framework of (Deng et al., 2025), denoted as **Deng-Distorted**, **Deng-ROI**, and **Deng-CostScaled**, respectively.

Since these baseline mechanisms do not incorporate budget constraints, we adopt the following adaptation: after executing each mechanism to completion, we select the longest prefix of the returned solution that satisfies the budget constraint. Given that all these mechanisms are greedy-based, this truncation strategy effectively captures the high-value portion of their solution while satisfying the budget. All baselines are implemented with the acceleration techniques

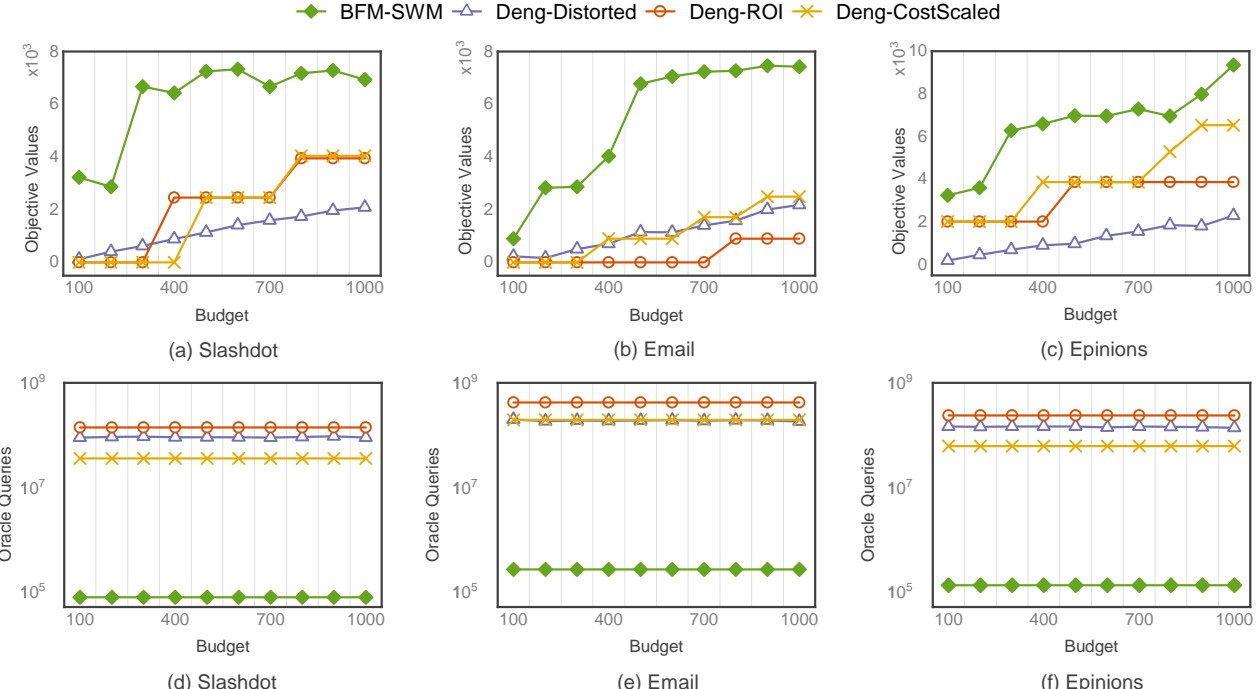

*Figure 1.* Experiments on Influence Maximization. The y-axis is on a logarithmic scale.

described in (Deng et al., 2025) to ensure efficient runtime performance. For our mechanisms, the structural parameters ($\alpha$, $\beta$ and $\ell$) are set according to the values derived from our theoretical analysis to maximize approximation ratios. Separately, for all implemented mechanisms, the error parameter $\epsilon$ (when applicable) is set uniformly to the default value of $0.1$. All experiments were conducted on a Windows workstation with an Intel Core Ultra 9 285K @ 3.70 GHz CPU[2].

### 6.2. Applications

We consider an application of influence maximization on social networks, which has been widely studied in prior work such as (Deng et al., 2025; Dütting et al., 2022; El Halabi et al., 2023; Han et al., 2023; El Halabi et al., 2024). In this application, given a social network graph $G = (\mathcal{N}, E)$, a buyer (e.g., an advertiser) aims to incentivize a subset of users $S \subseteq \mathcal{N}$ to spread information to their neighbors. Each user $u \in \mathcal{N}$ has a private cost $c(u)$ for providing the service. The buyer's valuation for the influence of a selected set $S$ is measured by the well-known coverage function $v(S) = \left| \bigcup_{u \in S} \mathcal{T}(u) \right|$, where $\mathcal{T}(u) = \{v \mid (u, v) \in E\}$ denotes the neighbor set of user $u$. It is well-established that this valuation function is submodular (Singer, 2010;

Dütting et al., 2022; El Halabi et al., 2023). The total payment made by the buyer to the selected users is constrained by a fixed budget $B$. Our objective is to maximize the social welfare, i.e., $v(S) - c(S)$, subject to this budget constraint. Our experiments are conducted on three large-scale network datasets from SNAP (Leskovec & Krevl, 2014): (1) Slashdot, with 77,360 nodes and 905,468 edges; (2) Email, with 265,214 nodes and 420,045 edges, and (3) Epinions, with 131,828 nodes and 841,372 edges.

### 6.3. Experimental Results

In this section, we evaluate the performance of all mechanisms based on two key metrics: (1) the *objective function value* (i.e., social welfare $v(S) - c(S)$), and (2) the *query complexity*, measured by the number of oracle queries to the valuation function $v(\cdot)$, which is hardware-independent and serves as the standard measure for computational efficiency in the value oracle model. We also record the wall-clock running time for all experiments. These results are deferred to Appendix D due to space constraints and further confirm the superior computational efficiency of our approach.

The results regarding objective function value are presented in Figure 1(a)–(c). It can be observed that our BFM-SWM mechanism consistently outperforms all baselines with significant advantages across the three datasets. Quantitatively, BFM-SWM achieves a social welfare ranging

---

[2]The source code is available at a repository: https://github.com/xue74193-dot/BFM-SWM.

from $1.22\times$ to $26.41\times$ that of the best-performing baseline at each budget level, with an overall average improvement factor of $4.49\times$. This substantial gain stems from the fact that our mechanism offers provable approximation guarantees under budget constraints, whereas the baseline mechanisms lack such theoretical guarantees. Furthermore, our mechanism also demonstrates significant advantages in computational efficiency, as evidenced by the oracle query results presented in Figure 1(d)–(f).

## 7. Conclusion

In this paper, we study the problem of mechanism design for submodular welfare maximization in procurement auctions. We propose BFM-SWM, the first budget-feasible mechanism with provable approximation guarantees for this objective. Furthermore, we develop a variant, BFM-VM, tailored for valuation maximization, which establishes a new state-of-the-art deterministic approximation ratio. The efficiency and effectiveness of our mechanisms are demonstrated both theoretically and experimentally.

Future work includes exploring broader settings, such as Nash welfare objectives and XOS/subadditive valuations in budget-feasible procurement. While some components of our framework may carry over, extending our theoretical analysis beyond submodular valuations appears to require new techniques. Establishing lower bounds for budget-feasible mechanisms with general submodular welfare objectives is another compelling open problem.

## Acknowledgments

The work of Shuang Cui was supported in part by the National Natural Science Foundation of China under Grant No. 62502333, and the Basic Research Program of Jiangsu under Grant No. BK20250786. The work of Yu-e Sun and He Huang was supported in part by the National Natural Science Foundation of China under Grant No. 62332013.

## Impact Statement

This paper presents work whose goal is to advance the field of machine learning. There are many potential societal consequences of our work, none of which we feel must be specifically highlighted here.

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

# Appendices

## A. Omitted Proofs

### A.1. Proof of Theorem 4.1

*Proof.* Due to the price rule in Line 11, we must have $\frac{v(u|S_{i,t}^u)}{\beta+\rho_t/B} \geq p(u)$ for any $u \in S_{i,t}$, which implies that $v(u \mid S_{i,t}^u) - \beta \cdot p(u) \geq \frac{p(u)\rho_t}{B}$. Moreover, considering the break condition in Line 13, the fact that $\beta > 1$ and $\rho_t > 0$, we can derive

$$\rho_t \geq v(S_{i,t}) - p(S_{i,t}) = \sum_{u \in S_{i,t}} v(u \mid S_{i,t}^u) - p(u)$$

$$\geq \sum_{u \in S_{i,t}} \frac{p(u)\rho_t}{B} = p(S_{i,t})\rho_t/B > 0,$$

which implies that the mechanism is budget-feasible (i.e., $p(S^*) \leq B$) and guarantees a non-negative surplus for the auctioneer.

The properties of obvious strategyproofness and individual rationality follow immediately from the observation that the mechanism takes the form of a descending clock auction (Milgrom & Segal, 2020).

Finally, we analyze the time complexity. We will show that the number of outer loop iterations $M$ in Algorithm 1 is bounded by $2 + \lceil \log_\alpha \frac{2OPT}{\epsilon} \rceil = \mathcal{O}(\log_\alpha \frac{OPT}{\epsilon})$. In round $M - 1$, at least one of the candidate solutions $\{S_{i,M-1}\}_{i=1}^\ell$, must have triggered the break instruction in Line 13 upon the evaluation of some element. Otherwise, we would have $R \setminus \left( \bigcup_{i=1}^\ell (S_{i,M-2} \cup S_{i,M-1}) \cup u^* \right) = \emptyset$, which would imply that round $M$ is never reached, contradicting the definition of $M$. Let $S'$ and $u'$ denote this specific candidate solution and the corresponding element, respectively. We proceed by contradiction. Suppose that $M > 2 + \lceil \log_\alpha \frac{2OPT}{\epsilon} \rceil$. It follows that

$$2 \left( v(O) - c(O) \right) \geq v(S') - c(S') + v(u') - c(u')$$

$$\geq v(S' \cup \{u'\}) - p(S' \cup \{u'\}) > \rho_{M-1} = \rho_1 \cdot \alpha^{M-2}$$

$$> 2OPT,$$

where the second inequality follows from the submodularity of $v(\cdot)$ and the fact that every element $u$ accepting the offered price at Line 12 satisfies $c(u) \leq p(u)$; the third inequality results from the condition triggering the break statement at Line 13; and the last inequality holds because $\rho_1 = \epsilon$. This yields a contradiction. Combining this with the fact that the inner loop executes at most $\mathcal{O}(n)$ times, the proof is complete. $\square$

### A.2. Proof of Lemma 4.3

*Proof.* Let $U_i = \bigcup_{t=1}^M S_{i,t}$ for $i \in \{1, 2\}$. By the owner rule in Algorithm 1, we have $U_1 \cap U_2 = \emptyset$. Using Definition 4.2, and the non-negativity and submodularity of $v(\cdot)$, we have

$$v(U_1 \cup O) - v(U_1) - \beta \cdot c(O \setminus U_1)$$

$$\leq \sum_{t=1}^M \sum_{u \in O_{1,t}^< \cup O_{2,t}^< \cup O_{2,t}^>} (v(u \mid U_1) - \beta \cdot c(u)) + v(u^*).$$

Here, the contribution of the final singleton class $O^*$ is upper bounded by $v(u^*)$, since $O^* \subseteq u^*$ and $v(\cdot)$ is nonnegative.

We now bound the right-hand side by analyzing the rejected elements and the elements first added to the other aggregate candidate sequence separately.

**Elements in $O_{1,t}^< \cup O_{2,t}^<$.** For elements in $O_{1,t}^<$, since $S_{1,t}^u \subseteq U_1$, submodularity gives $v(u \mid U_1) \leq v(u \mid S_{1,t}^u)$. For elements in $O_{2,t}^<$, by definition, such elements have not been successfully inserted into either aggregate candidate sequence before they reject. Hence, they are unassigned when they are considered in round $t$. Therefore, if such an element is routed to sequence 2, the greedy rule in Line 9 implies $v(u \mid S_{1,t}^u) \leq v(u \mid S_{2,t}^u)$. Combining this with $S_{1,t}^u \subseteq U_1$ and submodularity

gives $v(u \mid U_1) \le v(u \mid S_{2,t}^u)$. Thus,

$$\sum_{t=1}^{M} \sum_{u \in O_{1,t}^{<} \cup O_{2,t}^{<}} \left( v(u \mid U_1) - \beta \cdot c(u) \right)$$

$$\le \sum_{t=1}^{M} \left( \sum_{u \in O_{1,t}^{<}} \left( v(u \mid S_{1,t}^u) - \beta \cdot c(u) \right) + \sum_{u \in O_{2,t}^{<}} \left( v(u \mid S_{2,t}^u) - \beta \cdot c(u) \right) \right).$$

Consider any rejected element $u \in O_{1,t}^{<} \cup O_{2,t}^{<}$. Suppose that $u$ is routed to sequence $i \in \{1, 2\}$ in round $t$. The price $p(u)$ offered in round $t$ when $u$ exits the auction must be the lowest among all offers $u$ received, since prices are non-increasing. Since $u$ rejected this offer, we have $c(u) > p(u)$. Recall that the price is set as $p(u) = \min \left\{ p_{old}, \frac{v(u \mid S_{i,t}^u)}{\beta + \rho_t / B} \right\}$. Thus, the rejection implies $c(u) > \frac{v(u \mid S_{i,t}^u)}{\beta + \rho_t / B}$, which can be rearranged to $v(u \mid S_{i,t}^u) - \beta \cdot c(u) < \frac{\rho_t}{B} c(u)$. Substituting this back into the summation, we obtain

$$\sum_{t=1}^{M} \sum_{u \in O_{1,t}^{<} \cup O_{2,t}^{<}} \left( v(u \mid U_1) - \beta \cdot c(u) \right) \le \sum_{t=1}^{M} \sum_{u \in O_{1,t}^{<} \cup O_{2,t}^{<}} \frac{\rho_t}{B} c(u)$$

$$\le \sum_{t=1}^{M} \sum_{u \in O_{1,t}^{<} \cup O_{2,t}^{<}} \frac{\rho_M}{B} c(u) \le \frac{\rho_M}{B} \cdot c(O) \le \rho_M,$$

where the last inequality follows from $O$ being a feasible solution, i.e., $c(O) \le B$, and $\rho_t$ being non-decreasing.

**Elements in $O_{2,t}^{>}$.** For any $u \in O_{2,t}^{>}$, round $t$ is the first round in which $u$ is successfully inserted into the aggregate sequence $U_2$. Hence, $u$ was unassigned before this insertion, and the greedy rule in Line 9 chose sequence 2 in that round. Therefore, $v(u \mid S_{1,t}^u) \le v(u \mid S_{2,t}^u)$. Moreover, since $S_{1,t}^u \subseteq U_1$ and $u \notin U_1$, submodularity gives $v(u \mid U_1) \le v(u \mid S_{1,t}^u)$. Thus,

$$\sum_{t=1}^{M} \sum_{u \in O_{2,t}^{>}} \left( v(u \mid U_1) - \beta \cdot c(u) \right)$$

$$\le \sum_{t=1}^{M} \sum_{u \in O_{2,t}^{>}} \left( v(u \mid S_{2,t}^u) - \beta \cdot c(u) \right)$$

$$\le \sum_{t=1}^{M} \sum_{u \in S_{2,t}} \left( v(u \mid S_{2,t}^u) - \beta \cdot c(u) \right)$$

$$= \sum_{t=1}^{M} \left( v(S_{2,t}) - \beta \cdot c(S_{2,t}) \right).$$

The second inequality holds because $O_{2,t}^{>} \subseteq S_{2,t}$ and every element $u$ added to $S_{2,t}$ satisfies the acceptance condition $c(u) \le p(u) \le \frac{v(u \mid S_{2,t}^u)}{\beta + \rho_t / B}$, which implies $v(u \mid S_{2,t}^u) - \beta \cdot c(u) \ge \frac{\rho_t}{B} c(u) \ge 0$.

Combining the above two bounds, we obtain

$$v(U_1 \cup O) - v(U_1) - \beta \cdot c(O \setminus U_1)$$

$$\le \sum_{t=1}^{M} \left( v(S_{2,t}) - \beta \cdot c(S_{2,t}) \right) + \rho_M + v(u^*).$$

By symmetry, we similarly have

$$v(U_2 \cup O) - v(U_2) - \beta \cdot c(O \setminus U_2)$$

$$\le \sum_{t=1}^{M} \left( v(S_{1,t}) - \beta \cdot c(S_{1,t}) \right) + \rho_M + v(u^*).$$

Summing the above two inequalities gives

$$v(U_1 \cup O) + v(U_2 \cup O) - v(U_1) - v(U_2)$$
$$- \beta \cdot c(O \setminus U_1) - \beta \cdot c(O \setminus U_2)$$
$$\leq \sum_{t=1}^{M} v(S_{1,t}) + \sum_{t=1}^{M} v(S_{2,t}) + 2\rho_M + 2v(u^*).$$

In the last inequality, we have dropped the non-positive terms $-\beta \sum_{t=1}^{M} c(S_{1,t}) - \beta \sum_{t=1}^{M} c(S_{2,t})$. Since $U_1 \cap U_2 = \emptyset$, submodularity and non-negativity of $v(\cdot)$ imply

$$v(U_1 \cup O) + v(U_2 \cup O) \geq v(U_1 \cup U_2 \cup O) + v(O)$$
$$\geq v(O).$$

Moreover, since $c(\cdot)$ is nonnegative, we have

$$c(O \setminus U_1) + c(O \setminus U_2) \leq 2c(O).$$

Therefore,

$$v(O) - v(U_1) - v(U_2) - 2\beta c(O)$$
$$\leq \sum_{t=1}^{M} v(S_{1,t}) + \sum_{t=1}^{M} v(S_{2,t}) + 2\rho_M + 2v(u^*).$$

Finally, by subadditivity of nonnegative submodular functions, $v(U_1) + v(U_2) \leq \sum_{i=1}^{\ell} \sum_{t=1}^{M} v(S_{i,t})$. Combining the last two inequalities yields $v(O) - 2\beta \cdot c(O) \leq 2\rho_M + 2\sum_{i=1}^{\ell} \sum_{t=1}^{M} v(S_{i,t}) + 2v(u^*)$, which completes the proof. $\square$

### A.3. Proof of Lemma 4.7

*Proof.* By Lemma 4.5–4.6, Line 13 of Algorithm 1 and $S^* \leftarrow \mathrm{argmax}_{A \in \{S_{i,t} | i \in [\ell], t \in \{M-1, M\}\} \cup \{u^*\}} \{v(A) - p(A)\}$, we have for any $i \in [\ell]$

$$\sum_{t=1}^{M} v(S_{i,t}) \leq \sum_{t=M-1}^{M} \frac{v(S_{i,t}) - p(S_{i,t})}{1 - 1/\beta} + \sum_{t=1}^{M-2} \frac{v(S_{i,t}) - p(S_{i,t})}{1 - 1/\beta}$$
$$\leq \frac{2(v(S^*) - p(S^*))}{1 - 1/\beta} + \sum_{t=1}^{M-2} \frac{\rho_t}{1 - 1/\beta}$$
$$= \frac{2(v(S^*) - p(S^*))}{1 - 1/\beta} + \frac{\rho_{M-1} - \rho_1}{(\alpha - 1)(1 - 1/\beta)}$$
$$\leq \frac{2(v(S^*) - p(S^*))}{1 - 1/\beta} + \frac{2(v(S^*) - p(S^*))}{(\alpha - 1)(1 - 1/\beta)}$$
$$\leq \frac{2\alpha(v(S^*) - p(S^*))}{(\alpha - 1)(1 - 1/\beta)}.$$

The lemma then follows by summing the inequality over all $i \in [\ell]$. $\square$

### A.4. Proof of Theorem 4.8

*Proof.* We analyze the approximation ratio by considering two cases based on the total number of rounds $M$.

**Case 1:** $M \geq 2$. Combining Lemmas 4.3–4.7, using $v(u^*) - p(u^*) \leq v(S^*) - p(S^*)$ and rearranging the terms, we verify

that

$$v(O) - 2\beta \cdot c(O)$$

$$\leq 4\alpha \left( v(S^*) - p(S^*) \right) + \frac{4\ell \cdot \alpha \left( v(S^*) - p(S^*) \right)}{(\alpha - 1)(1 - 1/\beta)}$$

$$+ \frac{2}{1 - 1/\beta} (v(S^*) - p(S^*))$$

$$\leq \left( 4\alpha + \frac{10\alpha - 2}{(\alpha - 1)(1 - 1/\beta)} \right) (v(S^*) - c(S^*)),$$

where the final inequality is due to the observation that every element $u \in S^*$ must have accepted the price offered at Line 12, which implies $c(u) \leq p(u)$. Let $C = 4\alpha + \frac{10\alpha - 2}{(\alpha - 1)(1 - 1/\beta)}$. Rearranging the inequality yields

$$v(S^*) - c(S^*) \geq \frac{v(O)}{C} - \frac{2\beta}{C} c(O).$$

**Case 2:** $M \leq 1$. By Lemma 4.4, 4.6 and the definition of $S^*$, we have

$$v(O) - 2\beta \cdot c(O) \leq 2\epsilon + 6 \frac{v(S^*) - p(S^*)}{1 - 1/\beta}.$$

Rearranging terms and using $c(S^*) \leq p(S^*)$ gives:

$$v(S^*) - c(S^*) \geq \frac{1 - 1/\beta}{6} v(O) - \frac{\beta - 1}{3} c(O) - \frac{\epsilon(1 - 1/\beta)}{3}.$$

Combining the above two cases and setting $\alpha = 1 + \frac{2\sqrt{6}}{3}$ and $\beta = 4$ to optimize the approximation ratio, we finally get $v(S^*) - c(S^*) \geq \frac{3}{4(13 + 4\sqrt{6})} v(O) - c(O) - \epsilon/4 \geq 0.0328 \cdot v(O) - c(O) - \epsilon/4$, which completes the proof. $\qquad \square$

### A.5. Proof of Theorem 4.10

*Proof.* Similar to Theorem 4.8, we analyze the approximation ratio by considering two cases based on the total number of rounds $M$.

**Case 1:** $M \geq 2$. Combining Lemmas 4.5, 4.7, 4.9, using $v(u^*) - p(u^*) \leq v(S^*) - p(S^*)$ and rearranging the terms, we verify that

$$v(O) - \beta \cdot c(O)$$

$$\leq 2\alpha \left( v(S^*) - p(S^*) \right) + \frac{2\alpha \left( v(S^*) - p(S^*) \right)}{(\alpha - 1)(1 - 1/\beta)}$$

$$+ \frac{1}{1 - 1/\beta} (v(S^*) - p(S^*))$$

$$\leq \left( 2\alpha + \frac{3\alpha - 1}{(\alpha - 1)(1 - 1/\beta)} \right) (v(S^*) - c(S^*)),$$

where the final inequality is due to the observation that every element $u \in S^*$ must have accepted the price offered at Line 12, which implies $c(u) \leq p(u)$. Let $C = 2\alpha + \frac{3\alpha - 1}{(\alpha - 1)(1 - 1/\beta)}$. Rearranging the inequality yields

$$v(S^*) - c(S^*) \geq \frac{v(O)}{C} - \frac{\beta}{C} c(O).$$

**Case 2:** $M \leq 1$. By Lemma 4.6, 4.9, $\rho_1 = \epsilon$ and the definition of $S^*$, we have

$$v(O) - \beta \cdot c(O) \leq \epsilon + 2 \frac{v(S^*) - c(S^*)}{1 - 1/\beta}$$

Rearranging terms gives:

$$v(S^*) - c(S^*) \geq \frac{1 - 1/\beta}{2} v(O) - \frac{\beta - 1}{2} c(O) - \frac{\epsilon(1 - 1/\beta)}{2}.$$

Combining the above two cases and setting $\alpha = 1 + \frac{\sqrt{6}}{2}$ and $\beta = 3$ to optimize the approximation ratio, we finally get $v(S^*) - c(S^*) \geq 2v(O)/(13 + 4\sqrt{6}) - c(O) - \epsilon/3 \geq 0.0877 \cdot v(O) - c(O) - \epsilon/3$, which completes the proof. □

### A.6. Proof of Lemma 5.1

*Proof.* Using Definition 4.2 and the submodularity of $v(\cdot)$, we have

$$v(U_1 \cup O) - v(U_1) \leq \sum_{t=1}^{M} \sum_{u \in O_{1,t}^< \cup O_{2,t}^< \cup O_{2,t}^>} v(u \mid U_1).$$

For elements in $O_{1,t}^<$, since $S_{1,t}^u \subseteq U_1$, submodularity gives $v(u \mid U_1) \leq v(u \mid S_{1,t}^u)$. For elements in $O_{2,t}^<$, by definition, such elements have not been successfully inserted into either aggregate candidate sequence before they reject. Hence, they are unassigned when they are considered in round $t$. Therefore, if such an element is routed to sequence 2, the greedy rule in Line 9 implies $v(u \mid S_{1,t}^u) \leq v(u \mid S_{2,t}^u)$. Combining this with $S_{1,t}^u \subseteq U_1$ and submodularity gives $v(u \mid U_1) \leq v(u \mid S_{2,t}^u)$. Thus,

$$\sum_{t=1}^{M} \sum_{u \in O_{1,t}^< \cup O_{2,t}^<} v(u \mid U_1) \leq \sum_{t=1}^{M} \left( \sum_{u \in O_{1,t}^<} v(u \mid S_{1,t}^u) + \sum_{u \in O_{2,t}^<} v(u \mid S_{2,t}^u) \right).$$

Consider any rejected element $u \in O_{1,t}^< \cup O_{2,t}^<$. Suppose that $u$ is routed to sequence $i \in \{1, 2\}$ in round $t$. The price $p(u)$ offered in round $t$ when $u$ exits the auction must be the lowest among all offers $u$ received, since prices are non-increasing. Since $u$ rejected this offer, we have $c(u) > p(u)$. Recall that the price is set as $p(u) = \min \left\{ p_{old}, \frac{v(u \mid S_{i,t}^u)}{\rho_t / B} \right\}$. Thus, the rejection implies $c(u) > \frac{v(u \mid S_{i,t}^u)}{\rho_t / B}$, which can be rearranged to $v(u \mid S_{i,t}^u) < \frac{\rho_t}{B} c(u)$. Substituting this back into the summation, we obtain

$$\sum_{t=1}^{M} \sum_{u \in O_{1,t}^< \cup O_{2,t}^<} v(u \mid U_1) \leq \sum_{t=1}^{M} \sum_{u \in O_{1,t}^< \cup O_{2,t}^<} \frac{\rho_t}{B} c(u)$$

$$\leq \sum_{t=1}^{M} \sum_{u \in O_{1,t}^< \cup O_{2,t}^<} \frac{\rho_M}{B} c(u) \leq \frac{\rho_M}{B} \cdot c(O) \leq \rho_M,$$

where the last inequality follows from $O$ being feasible, i.e., $c(O) \leq B$, and $\rho_t$ being non-decreasing.

For any $u \in O_{2,t}^>$, round $t$ is the first round in which $u$ is successfully inserted into the aggregate sequence $U_2$. Hence, $u$ was unassigned before this insertion, and the greedy rule in Line 9 chose sequence 2 in that round. Therefore, $v(u \mid S_{1,t}^u) \leq v(u \mid S_{2,t}^u)$. Moreover, since $S_{1,t}^u \subseteq U_1$ and $u \notin U_1$, submodularity gives $v(u \mid U_1) \leq v(u \mid S_{1,t}^u)$. Thus,

$$\sum_{t=1}^{M} \sum_{u \in O_{2,t}^>} v(u \mid U_1) \leq \sum_{t=1}^{M} \sum_{u \in O_{2,t}^>} v(u \mid S_{2,t}^u)$$

$$\leq \sum_{t=1}^{M} \sum_{u \in S_{2,t}} v(u \mid S_{2,t}^u) = \sum_{t=1}^{M} v(S_{2,t}).$$

The second inequality holds because $O_{2,t}^> \subseteq S_{2,t}$ and every element $u$ added to $S_{2,t}$ satisfies the acceptance condition $c(u) \leq p(u) \leq \frac{v(u \mid S_{2,t}^u)}{\rho_t / B}$, which implies $v(u \mid S_{2,t}^u) \geq \frac{\rho_t}{B} c(u) \geq 0$.

Combining the above bounds, we obtain $v(U_1 \cup O) - v(U_1) \le \sum_{t=1}^{M} v(S_{2,t}) + \rho_M$. By symmetry, we similarly have $v(U_2 \cup O) - v(U_2) \le \sum_{t=1}^{M} v(S_{1,t}) + \rho_M$.

Summing the two inequalities and using $U_1 \cap U_2 = \emptyset$, submodularity, and non-negativity of $v(\cdot)$ completes the proof.

$\square$

### A.7. Proof of Lemma 5.2

*Proof.* Consider round $M - 1$. At least one of the candidate sets $S_{1,M-1}$ or $S_{2,M-1}$ must have triggered the break instruction upon the evaluation of some element; otherwise, we would have $R \setminus \left( \bigcup_{i=1}^{\ell} (S_{i,M-2} \cup S_{i,M-1}) \right) = \emptyset$, which would imply that round $M$ is never reached, contradicting the definition of $M$. Let $S'$ denote such a candidate solution and let $u'$ be the corresponding element whose evaluation triggers the break condition. Using the submodularity of $v(\cdot)$ together with the condition that triggers the break instruction in Line 13 (i.e., $v(S' \cup \{u'\}) > \rho_{M-1}$), we obtain $\rho_{M-1} \le v(S' \cup \{u'\}) \le v(S') + v(u')$. Since $\rho_1$ is initialized as $\max_{u \in R} \{v(u)\}$, it follows that $v(u') \le \rho_1$. Thus, we obtain $v(S') + \rho_1 \ge \rho_{M-1} = \frac{\rho_M}{\alpha}$.

Rearranging the inequality yields the following cases:

$$
\begin{aligned}
M \ge 3 &: v(S') \ge \rho_M/\alpha - \rho_1 \ge \rho_M/\alpha - \rho_M/\alpha^2 \\
M = 2 &: v(S') = \rho_1 = \rho_M/\alpha \\
M = 1 &: v(S') = \rho_1 = \rho_M
\end{aligned}
$$

In all cases, since $\alpha > 1$, we have $\frac{\alpha^2}{\alpha-1} v(S') \ge \rho_M$. The lemma then follows from the definition of $S^*$ in Line 26. $\square$

### A.8. Proof of Lemma 5.3

*Proof.* Using Lemma 5.2 together with the break condition in Line 13, for any $i \in [\ell]$ we have

$$
\begin{aligned}
\sum_{t=1}^{M} v(S_{i,t}) &\le v(S_{i,M}) + v(S_{i,M-1}) + \sum_{t=1}^{M-2} v(S_{i,t}) \\
&\le v(S_{i,M}) + v(S_{i,M-1}) + \sum_{t=1}^{M-2} \rho_t \\
&\le v(S_{i,M}) + v(S_{i,M-1}) + \frac{\rho_{M-1} - \rho_1}{\alpha - 1} \\
&\le v(S_{i,M}) + v(S_{i,M-1}) + \frac{v(S^*)}{\alpha - 1}.
\end{aligned}
$$

Using the definition of $S^*$ completes the proof. $\square$

### A.9. Proof of Theorem 5.4

*Proof.* Combining Lemmas 5.1–5.3 and rearranging terms, we obtain $v(S^*) \ge \frac{\alpha-1}{2(\alpha^2+4\alpha-2)} v(O)$. Setting $\alpha = 1 + \sqrt{3}$ to optimize the approximation ratio, we can get $v(S^*) \ge v(O)/(12 + 4\sqrt{3})$.

Then we analyze the time complexity. We will show that the number of outer loop iterations $M$ in Algorithm 2 is bounded by $2 + \lceil \log_\alpha n + \log_\alpha 2 \rceil = \mathcal{O}(\log n)$. Let $S'$ and $u'$ denote the candidate solution and element, respectively, whose evaluation triggers the break condition in Line 13 during round $M - 1$. We proceed by contradiction. Suppose that $M > 2 + \lceil \log_\alpha n + \log_\alpha 2 \rceil$. It follows that

$$
2v(O) \ge v(S') + v(u') \ge v(S' \cup \{u'\}) \ge \rho_{M-1} = \rho_1 \cdot \alpha^{M-2} > 2n\rho_1 \ge 2v(O),
$$

where the second inequality follows from the submodularity of $v(\cdot)$; the third inequality results from the condition triggering the break statement at Line 13; and the last inequality holds because $\rho_1 = \max_{u \in R} v(u)$. This yields a contradiction. Combining this with the fact that the inner loop executes at most $\mathcal{O}(n)$ times, the proof is complete. $\square$

---

**Algorithm 2** BFM-VM: A Deterministic Budget-Feasible Mechanism for Submodular Valuation Maximization

---

**input** budget $B$, parameters $\alpha > 1$, and number of candidate set sequences $\ell \in \{1, 2\}$

1: Let $R \leftarrow \{u \in \mathcal{N} \mid u \text{ accepts price } B\}$ and $p(u) \leftarrow B$ for each $u \in R$

2: Initialize $t \leftarrow 1$; $\rho_t \leftarrow \max_{u \in R} v(u)$; $S_{i,t} \leftarrow \emptyset$ for each $i \in [\ell]$; let $a \in \arg\max_{u \in R} v(u)$; $S_{1,t} \leftarrow \{a\}$; $\text{owner}(u) \leftarrow 0$ for each $u \in R$ and $\text{owner}(a) \leftarrow 1$

3: **repeat**

4:     $t \leftarrow t + 1$; $\rho_t \leftarrow \alpha \cdot \rho_{t-1}$; $S_{i,t} \leftarrow \emptyset$ for all $i \in [\ell]$

5:     **for** $u \in R \setminus \left( \bigcup_{i=1}^{\ell} S_{i,t-1} \right)$ **do**

6:         **if** $\text{owner}(u) \neq 0$ **then**

7:             $j \leftarrow \text{owner}(u)$

8:         **else**

9:             $j \leftarrow \arg\max_{i \in [\ell]} v(u \mid S_{i,t})$

10:         **end if**

11:         Update $p(u) \leftarrow \min\left\{ p(u), \frac{v(u \mid S_{j,t})}{\rho_t / B} \right\}$

12:         **if** $u$ accepts price $p(u)$ **then**

13:             **if** $v(S_{j,t} \cup \{u\}) > \rho_t$ **then**

14:                 **break**

15:             **else**

16:                 $S_{j,t} \leftarrow S_{j,t} \cup \{u\}$;

17:                 **if** $\text{owner}(u) = 0$ **then**

18:                     $\text{owner}(u) \leftarrow j$

19:                 **end if**

20:             **end if**

21:         **else**

22:             $R \leftarrow R \setminus \{u\}$;

23:         **end if**

24:     **end for**

25: **until** $R \setminus \left( \bigcup_{i=1}^{\ell} (S_{i,t-1} \cup S_{i,t}) \right) = \emptyset$

26: Set $M \leftarrow t$; $S^* \leftarrow \arg\max_{A \in \{S_{i,t} \mid i \in [\ell], t \in \{M-1, M\}\}} v(A)$

**output** $S^*$ and $p(u)$ for each $u \in S^*$

---

# B. Pseudocode of the BFM-VM Mechanism

# C. Experiments on Valuation Maximization

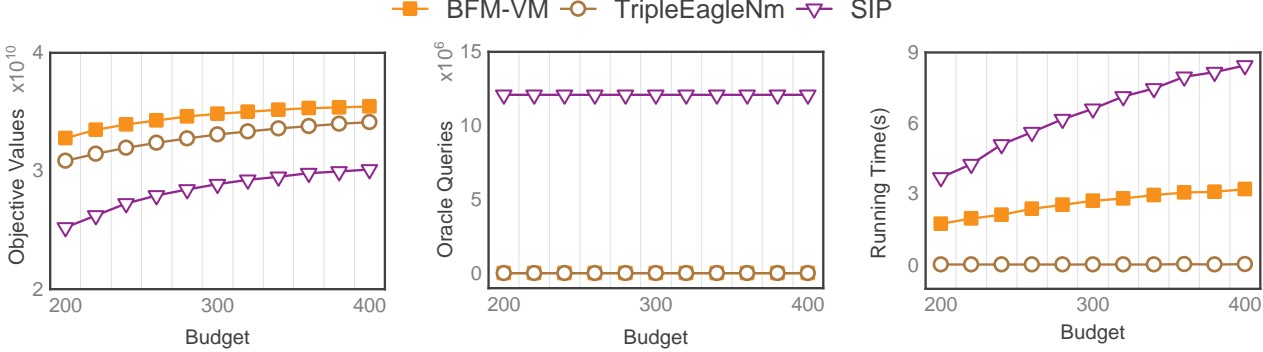

*Figure 2.* Experiments on the Crowdsourcing Application

In this section, we conduct experiments on a crowdsourcing application similar to those in (Han et al., 2025; Huang et al., 2023; Jalaly & Tardos, 2021). In this setting, the set of workers (sellers) $\mathcal{N}$ each possess an image, and a buyer seeks to crowdsource a representative and diverse set $S$ of images from the workers. Following (Han et al., 2025; Huang et al., 2023; Fahrbach et al., 2019; Mirzasoleiman et al., 2016), we use the CIFAR-10 dataset (Krizhevsky, 2009) that contains tens of thousands of $32 \times 32$ color images, and the buyer's valuation for a selected set $S$ is defined as: $v(S) = \frac{1}{|\mathcal{N}|} \left( \sum_{u \in \mathcal{N}} \sum_{v \in S} s_{u,v} - \sum_{u \in S} \sum_{v \in S} s_{u,v} \right)$, where $s_{u,v}$ denotes the similarity between images $u$ and $v$, quantified via the inner product of their 3,072-dimensional pixel vectors. As indicated by (Mirzasoleiman et al., 2016), this valuation function is non-monotone and submodular, where the first term and second term capture the coverage and diversity of $S$, respectively. Each worker $u$ has a private cost $c(u)$ for providing their image. Following (Mirzasoleiman et al., 2016; Han et al., 2025), we set the cost proportional to the standard deviation of the image's pixel intensities, normalized to have an average value of 0.1, which assigns higher costs to high-contrast images and lower costs to blurry images. The buyer operates under a budget constraint $B$ on the total payment to workers. Our objective is to maximize the valuation $v(S)$ subject to this budget constraint. Following (Han et al., 2025; Huang et al., 2023), we use images with labels in {Airplane, Automobile, Bird} as our groundset $\mathcal{N}$.

In this application, we compare our BFM-VM mechanism tailored for non-monotone submodular valuation maximization in procurement auctions, against state-of-the-art mechanisms for the same objective. The baselines include:

- **TripleEagleNm** (Han et al., 2025), which achieves the best-known randomized approximation ratio for non-monotone submodular valuation maximization.

- **Simultaneous-Iterative-Pruning, abbreviated as SIP** (Balkanski et al., 2022), which achieves the best-known deterministic approximation ratio for non-monotone submodular valuation maximization.

We evaluate the performance of all mechanisms based on three metrics: (1) the objective function value (i.e., valuation $v(S)$), (2) the number of oracle queries to the valuation function $v(\cdot)$, and (3) the wall-clock running time. The experimental results are summarized in Figure 2. It is evident that our by-product variant, BFM-VM, consistently surpasses all baselines in terms of objective function value. Quantitatively, BFM-VM achieves an average improvement of 21.85% over SIP and 5.27% over TripleEagleNm. Furthermore, regarding computational efficiency, our deterministic mechanism BFM-VM significantly outperforms SIP, the current state-of-the-art deterministic mechanism. These empirical results underscore the effectiveness of our approach when extended to valuation maximization objectives.

## D. Wall-Clock Running Time for Influence Maximization

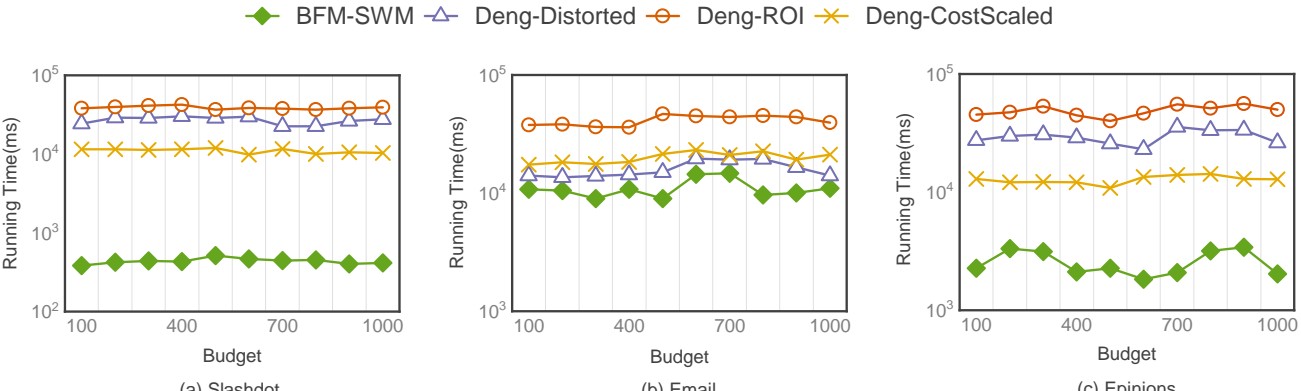

*Figure 3.* Comparison of wall-clock running time for influence maximization across three datasets.

In this section, we present the wall-clock running time results (Figure 3) for the welfare maximization experiments (influence maximization application) presented in Section 6. As noted in the main text, the observed running time trends are consistent with the query complexity results, further validating the superior computational efficiency of our BFM-SWM mechanism.

