# OpenReview forum: "Budget-Feasible Mechanisms for Submodular Welfare Maximization in Procurement Auctions"
_ICML.cc/2026/Conference — ICML 2026 regular_

### Official Review · Reviewer_Ce66 · 2026-03-11

**Soundness:** 3
**Presentation:** 2
**Significance:** 3
**Originality:** 3
**Overall Recommendation:** 4
**Confidence:** 5

**Summary:**

This paper studies the design of budget-feasible mechanisms that achieve provable approximation guarantees for welfare maximization in procurement auctions. The paper makes novel technical contribution by proposing the first such mechanism (BFM-SWM) and achieves a deterministic approximation ratio of 0.0328 for general submodular functions. The variant BFM-VM also establishes state-of-the-art results for the classical valuation maximization problem.

**Compliance With Llm Reviewing Policy:**

Affirmed.

**Final Justification:**

The author's response has effectively resolved my issue. I will maintain my positive score.

**Key Questions For Authors:**

Please refer to the weakness

**Limitations:**

yes

**Strengths And Weaknesses:**

Strength:
1. In the research on the budget-feasible mechanism, different from the traditional optimization objective (value maximization), this paper proposes the welfare maximization optimization objective, which is the total value obtained by the buyer minus the cost of the items.
2. For such optimization objectives, under the general submodular function, this paper proposes the truthful incentive mechanism and provides effective theoretical performance guarantees.
3. The author also simultaneously proves that although the proposed method optimizes social welfare, it can still be applied to valuation maximization, and also demonstrates that the approximation ratio in this case achieves a relatively large improvement compared to the best-known method.

Weakness:
1. My major concern is the motivation to optimize social welfare in the budget-feasible problem. The core issue lies in the interplay between the welfare objective and the budget constraint. When the mechanism cannot find positive-welfare items within budget, a common scenario in many real-world procurement settings, it returns an empty set, leaving the entire budget unspent. In critical public applications such as vaccine procurement, emergency response, or public infrastructure, leaving budget unspent while needs exist is politically and economically untenable. The paper's welfare objective implicitly assumes that "doing nothing" is preferable to "doing something with zero or slightly negative net welfare," which contradicts the operational realities of most procurement agencies with budgets. While the technical machinery is sophisticated and the results are novel within the stated framework, the problem formulation itself limits practical applicability. The author should further explain on the scenarios in practical applications, which will determine my future evaluation of this paper.
2. The paper does not explain why the weaker approximation notion was adopted. Is it because using the common approximation may lead to non-approximable results? If so, the authors need to provide some explanations of the impossibility results.
3. Theorem 4.8 gives the upper bound on performance under the optimization of social welfare, how about the lower bound?
4. In the experimental section, the selection of the baseline involves comparing the submodular optimization method, without making a comparison with the existing budget-feasible mechanism in the dimension of social welfare.

---

> ### Author Rebuttal · Authors · 2026-03-29
>
> Thanks for your positive assessment. We address your concerns below.
> > W1
>
> As noted by [1], the welfare maximization objective in procurement auctions can be viewed as analogous to the "gains from trade" concept in the bilateral trade literature, as it measures the net social value generated by running the auction. This objective promotes both cost efficiency and allocative efficiency, which are essential for sustainable procurement.
>
> More importantly, in broad commercial applications, when no positive-welfare items can be found within the budget, returning an empty set is not a flaw, but rather a crucial "stop-loss" mechanism and a valuable economic signal indicating fundamental market mismatch (i.e., sellers' true costs exceed the buyer's valuation). "Doing nothing" is the rational choice to prevent the buyer from running a deficit. Examples include:
>
> 1. Viral Marketing [2]: Advertisers pay to incentivize social network users as seeds. If the true seed costs exceed the expected revenue, running the campaign inevitably causes a net financial loss. Halting the campaign is the rational business decision.
>
> 2. Crowdsourcing Markets [3]: Requesters hire strategic workers for tasks under a budget. If workers' true costs exceed the value of completed tasks, rational requesters should hold their budget and wait for a more suitable labor supply rather than overpaying.
>
> In these commercial settings, prior Budget-Feasible Mechanisms (BFMs) that only maximize valuation may result in negative auctioneer surplus (a deficit). Our welfare objective and the non-negative auctioneer surplus property of our mechanism ensure sustainable, deficit-free procurement, a guarantee that no prior BFM offers.
>
> Moreover, we agree that in critical public applications (e.g., emergency response), leaving budget unspent is politically untenable. However, we respectfully note that in such life-saving scenarios, the societal valuation inherently reflects the extreme urgency and massive public benefit, naturally making it vastly exceed the procurement costs. So our mechanism still reliably procures items rather than returning an empty set. We also note that for settings where budget must be utilized regardless of net welfare, our framework accommodates this via the Valuation Maximization variant in Sec. 5.
>
> We'll revise to clarify these points.
>
> > W2
>
> As detailed in Lines 50-66 and Lines 170-179 of our paper, [4] proves that even in the simpler setting of pure submodular optimization (where costs are public and economic properties are not required), no polynomial-time algorithm can achieve a non-trivial multiplicative approximation ratio for maximizing a possibly negative submodular function $v-c$ (e.g., our welfare function). Thus, following extensive prior work on maximizing $v-c$ under public costs, we adopt a weaker approximation notion.
>
> > W3
>
> To our knowledge, even for the simpler problem of BFM design for general (not necessarily monotone) submodular valuation functions, no non-trivial approximation lower bound has been established yet (see the survey [Liu et al., IJCAI 2024] cited in our paper). Since our welfare maximization problem is more complex, establishing a lower bound is more challenging. Thus, while Theorem 4.8 provides the first provable approximation guarantee for the problem, establishing a theoretical lower bound remains a highly non-trivial open question. We consider this a compelling direction for future work.
>
> > W4
>
> To our knowledge, existing BFMs are designed for valuation maximization (VM) and offer no provable approximation ratios for social welfare maximization (SWM). [1] is the only existing work developing SWM mechanisms with provable approximation ratios, albeit without considering budget feasibility. [1] proposed a framework that converts regularized submodular maximization algorithms into mechanisms with economic properties. Thus, in our experiments on Influence Maximization (an SWM application), we compared our BFM-SWM against submodular optimization algorithms converted by their framework. In the appendix, we evaluated our BFM-VM against existing SOTA BFMs (including TripleEagleNm and SIP) on a VM application named Crowdsourcing.
>
> Following your advice, we conducted additional experiments comparing BFMs (TripleEagleNm and SIP) in the social welfare maximization application. Results confirm that our BFM-SWM consistently achieves a significant advantage in the objective function value, with performance gains of up to 97.47%. Given a chance, we'll include additional experiments.
>
> **References**
>
> [1] Procurement Auctions via Approximately Optimal Submodular Optimization. ICML
>
> [2] Unconstrained submodular maximization with modular costs:Tight approximation and application to profit maximization. PVLDB
>
> [3] Optimal selection of crowdsourcing workers balancing their utilities and platform profit. IOTJ
>
> [4] An efficient framework for balancing submodularity and cost. KDD

---

> > ### Author Rebuttal · Reviewer_Ce66 · 2026-04-01
> >
> > I appreciate the authors' response. Based on the current reply, I have no further questions and will maintain a positive score.

---

### Official Review · Reviewer_5bej · 2026-03-13

**Soundness:** 4
**Presentation:** 3
**Significance:** 4
**Originality:** 3
**Overall Recommendation:** 5
**Confidence:** 4

**Summary:**

The authors study the design of budget-feasible mechanism for submodular optimization in procurement auctions. Previous work focused on the valuation maximization or relaxed the budget-feasible constraint. In this paper, they proposed the first mechanism to achieve constant approximation for welfare maximization, maintaining the guarantees of truthfulness, individual rationality and budget feasibility. The main idea is to construct a geometrically increasing threshold rule in a descending clock auction framework. Besides, as a by-product, a variant of the main contribution improve both the ratio and running time for general submodular functions with the regime of valuation maximization.

**Compliance With Llm Reviewing Policy:**

Affirmed.

**Final Justification:**

I thank the authors for their detailed responses. The revisions and clarifications have addressed my concerns, and I am happy to maintain my positive score.

**Key Questions For Authors:**

1. Is there any lower bound for the ratio?
2. Do you think about ratio analysis of Nash welfare for this problem?

**Limitations:**

yes

**Strengths And Weaknesses:**

Strengths:
1. The authors study an important open problem in budget-feasible mechanism design.
2. They also improve the state-of-the-art ratio for the problem of valuation maximization.


Weaknesses:
1. Presentation should be improved for clarity.
2. The ratio is somehow small compared with that of the classic submodular optimization.
3. See questions.

---

> ### Author Rebuttal · Authors · 2026-03-29
>
> We sincerely thank you for your highly positive evaluation of our paper. We address your specific questions below.
>
> > The ratio is somehow small compared with that of the classic submodular optimization.
>
> **Response:** We would like to respectfully clarify the fundamental difference between classic submodular optimization and Budget-Feasible Mechanism (BFM) design that causes this gap in approximation ratios.
>
> In classic submodular optimization, the costs are public information and the goal is solely to maximize the objective. In contrast, in our procurement auction setting, the costs are private information held by strategic sellers, and BFMs must not only optimize the objective but also strictly guarantee some economic properties, such as truthfulness and individual rationality. Thus, BFM design is much harder than pure submodular optimization, inevitably leading to smaller approximation ratios. For instance, for general (not necessarily monotone) submodular valuation maximization under a knapsack constraint, the state-of-the-art optimization algorithm achieves a $0.401$-approximation in polynomial time, while the best-known randomized BFM achieves a $0.0833$-approximation, and the previous best deterministic BFM achieves a $0.0156$-approximation with an $O(n^2 \log n)$ running time (our work improves this deterministic ratio to $0.0528$ while reducing the running time to $O(n \log n)$). The submodular welfare objective $v(\cdot)-c(\cdot)$ is structurally more complex (a submodular valuation minus a modular cost function that may render the objective negative). There were no BFMs with provable approximation guarantees for this objective prior to our work, and our mechanisms achieve the first theoretical breakthrough for this problem.
>
>
> > Q1: Is there any lower bound for the ratio?
>
> **Response:** To our knowledge, even for the simpler problem of BFM design for general (not necessarily monotone) submodular valuation maximization, no non-trivial lower bound on the approximation ratio has been established yet (see the survey [Liu et al., IJCAI 2024] cited in our paper). Since our welfare maximization objective is more complex, analyzing the lower bound becomes more challenging. We consider this a compelling direction for future work.
>
> > Q2: Do you think about ratio analysis of Nash welfare for this problem?
>
> **Response:** We sincerely thank the reviewer for raising this thought-provoking question. Nash welfare is indeed an elegant objective that naturally balances economic efficiency and fairness among participating agents. In contrast, our current social welfare objective measures the net social value generated by the auction (analogous to "gains from trade" in the bilateral trade literature), promoting cost efficiency and allocative efficiency for sustainable procurement. However, the mathematical structure of Nash welfare (typically defined as the product or geometric mean of agents' utilities) differs from our welfare function, e.g., it inherently lacks the submodularity of our welfare function. Thus, extending our current framework to handle Nash welfare while preserving economic properties and provable approximation guarantees appears to be a different and non-trivial problem. We believe this is a brilliant direction for future research, and we will add a discussion of Nash welfare in the conclusion of the revised manuscript.

---

> > ### Author Rebuttal · Reviewer_5bej · 2026-04-06
> >
> > I thank the authors for their detailed responses. The revisions and clarifications have addressed my concerns, and I am happy to maintain my positive score.

---

### Official Review · Reviewer_Yd1C · 2026-03-13

**Soundness:** 4
**Presentation:** 4
**Significance:** 3
**Originality:** 4
**Overall Recommendation:** 6
**Confidence:** 4

**Summary:**

This paper studies budget-feasible mechanisms in procurement auctions for submodular welfare maximization. The authors propose a novel mechanism, BFM-SWM, a multi-round descending clock auction that satisfies obvious strategyproofness, individual rationality, non-negative auctioneer surplus, and budget feasibility. he mechanism achieves apporoximation guarantees of $0.0328$ and $0.0877$ for general and monotone submodular valuations, respectively. The authors also propose the BFM-VM for valuation maximization objective, which achieves an approximation ratio of $1/(12+4\sqrt{3})$, significantly improving the best known bound of $1/64$. Empirical results are also presented to support the theoretical findings.

**Compliance With Llm Reviewing Policy:**

Affirmed.

**Final Justification:**

Thank the authors for their effort. I support accepting the paper and look forward to seeing the final version!

**Key Questions For Authors:**

1. Can your techniques be extended to other valuation classes, such as XOS or subadditive functions? In particular, could you offer some intuition about which parts of the analysis would continue to hold and which would break down or require new ideas?

2. In the experiments, the query counts for BFM-SWM appear to remain nearly constant across different budget levels. Could you provide some explanation for this behavior?

**Limitations:**

N/A.

**Strengths And Weaknesses:**

1. This is the first paper to study budget-feasible mechanisms with provable approximation guarantees for submodular welfare maximization. The problem is both practically motivated and theoretically challenging.

2. The results and technical contributions are strong. The proposed mechanism is novel, and the analysis is highly nontrivial. Moreover, the variant mechanism BFM-VM significantly improves both the best known approximation bound and the running time.

3. The paper is well written, with clear motivation and a detailed presentation of the mechanism’s execution, as well as intuitive high-level explanations of the approximation-ratio analysis.

---

> ### Author Rebuttal · Authors · 2026-03-29
>
> We deeply appreciate your encouraging feedback and strong support for our work. We address your questions below.
>
> > Can your techniques be extended to other valuation classes, such as XOS or subadditive functions? In particular, could you offer some intuition about which parts of the analysis would continue to hold and which would break down or require new ideas?
>
> **Response:** We sincerely thank the reviewer for this insightful and forward-looking question. Extending our framework to broader valuation classes such as XOS or subadditive functions is an interesting yet highly non-trivial direction.
>
> To our knowledge, for XOS and subadditive valuations, no non-trivial approximation can be achieved using only a polynomial number of value queries; a demand oracle is typically required instead of our current value oracle. Moreover, since these classes lack submodularity, the parts of our analysis that rely on submodular properties would require new techniques.
>
> On the positive side, the clock-auction structure and the associated economic properties (e.g., truthfulness and individual rationality) should remain applicable, as they do not depend on the specific form of the objective. Similarly, our use of the parameter $\beta$ to control the value-to-payment ratio, and thereby guarantee non-negative auctioneer surplus, relies primarily on the pricing rule and is likely to carry over.
>
> We believe that designing budget-feasible mechanisms for XOS/subadditive welfare maximization represents a promising open problem. Given the opportunity, we will include a brief discussion of these intuitions in the revised manuscript.
>
> > In the experiments, the query counts for BFM-SWM appear to remain nearly constant across different budget levels. Could you provide some explanation for this behavior?
>
> **Response:** Thanks for your feedback. The seemingly constant query count for BFM-SWM is the combined result of a visual artifact from the logarithmic y-axis scale and the inherent efficiency of our algorithmic design.
>
> Empirically, the number of oracle queries does increase slowly as the budget grows. However, because the y-axis is plotted on a logarithmic scale that spans several orders of magnitude (to accommodate the much larger query counts of the baseline algorithms), this modest upward trend appears visually compressed into a nearly flat line.
>
> From an algorithmic perspective, the running time of BFM-SWM inherently exhibits a very weak dependence on the budget. Although a larger budget allows more items to be selected, the overall running time is $O(n\log \frac{OPT}{\epsilon})$ as shown by our theoretical analysis, which indicates that the number of queries is predominantly determined by the ground set size $n$, rather than scaling heavily with the budget.
>
> Given the opportunity, we will explicitly discuss this behavior in our revised paper.

---

> > ### Author Rebuttal · Reviewer_Yd1C · 2026-04-01
> >
> > I thank the authors for their reponses. I have no further questions and will maintain the positive score. Looking forward to seeing the revised version of the paper.

---

### Official Review · Reviewer_faA1 · 2026-03-13

**Soundness:** 3
**Presentation:** 3
**Significance:** 3
**Originality:** 3
**Overall Recommendation:** 4
**Confidence:** 2

**Summary:**

This paper studies **budget-feasible procurement mechanism** for a buyer with a **submodular valuation** over items owned by strategic sellers with private costs. Unlike most prior work in budget-feasible mechanism design, which focuses on maximizing valuation ($v(S)$), the paper targets **social welfare** ($v(S)-c(S)$).

The main contribution is a deterministic descending-clock mechanism (BFM-SWM) for submodular welfare maximization. The paper also derives, as a by-product, an improved deterministic mechanism for the classical valuation-maximization objective.

**Compliance With Llm Reviewing Policy:**

Affirmed.

**Key Questions For Authors:**

1) Could you please address the point made above? Is it a mistake or a mis-interpretation on my side?

2) In Section 1.2, you highlight the geometrically increasing threshold, the singleton candidate solution, and the parameter $\beta$ as core technical ingredients. Since geometrically increasing thresholds are already a standard tool in budget-feasible mechanism design and clock-auction style analyses, could you clarify more explicitly what is the technical novelty in your use and analysis of the threshold here relative to prior work? In particular, which part of the threshold analysis fundamentally relies on the welfare objective $v(S)-c(S)$ potentially being negative, rather than following an adaptation of existing budget-feasible valuation-maximization arguments?

**Limitations:**

yes

**Strengths And Weaknesses:**

Strengths:

- The paper addresses a very natural and well-motivated extension of budget-feasible mechanism design: maximizing welfare rather than just buyer valuation.

- The technical setting seems non-trivial: Because costs are private, the mechanism cannot directly optimize $(v(S)-c(S))$ and the welfare objective may be non-monotone or even negative. The paper develops a mechanism tailored to this difficulty rather than applying existing valuation-maximization tools in a black-box way.

- The proposed mechanism seems reasonably novel at a high-level.


Weaknesses:

- The claim that BFM-VM preserves “all economic properties” from Theorem 4.1 appears false, at least for non-negative auctioneer surplus. The paper states that after setting $\beta=0$, removing $u^\star$, changing the initialization, and selecting the final set by pure value, “the economic properties established in Theorem 4.1 obviously continue to hold.” However, consider two sellers with additive valuation $v(\{1\})=v(\{2\})=1$, costs $c_1=c_2=0$, budget $B=100$, and the parameter choice $\alpha=1+\sqrt{3}$ from Theorem 5.4. In Algorithm 2, line 1 initializes $p(1)=p(2)=100$, and line 2 initializes $S_{1,1}$ to a best singleton, say $\{1\}$. Since the final output is chosen by maximizing $v(A)$ rather than $v(A)-p(A)$, the mechanism can output $\{1\}$ with payment $100$, yielding auctioneer surplus $v(S^\star)-p(S^\star)=1-100<0$.

It seems that the surplus proof in Theorem 4.1 relies on $\beta>1$ through inequalities of the form $v(S_{i,t})-p(S_{i,t}) \ge p(S_{i,t})\rho_t/B$, and that argument disappears once $\beta=0$.

---

> ### Author Rebuttal · Authors · 2026-03-29
>
> Thank you for the positive assessment of our work. We address your questions below.
>
> > The claim that BFM-VM preserves "all economic properties" from Theorem 4.1 appears false, at least for non-negative auctioneer surplus...
>
> > Q1. Could you please address the point made above? Is it a mistake or a mis-interpretation on my side?
>
> **Response:** We sincerely thank you for the careful reading and the concrete counterexample. Your interpretation is correct, and we apologize for the imprecise claim in the paper. To clarify: our main BFM-SWM mechanism (for welfare maximization) does guarantee a non-negative auctioneer surplus; while our BFM-VM variant (for valuation maximization) does not, **consistent with all prior BFMs** in the literature. We will correct the statement in the revised manuscript to precisely reflect that BFM-VM preserves all economic properties established in Theorem 4.1 except the non-negative auctioneer surplus (namely, Truthfulness, Individual Rationality, and Budget Feasibility).
>
>
> > Q2. In Section 1.2, you highlight the geometrically increasing threshold, the singleton candidate solution, and the parameter $\beta$ as core technical ingredients. Since geometrically increasing thresholds are already a standard tool in budget-feasible mechanism design and clock-auction style analyses, could you clarify more explicitly what is the technical novelty in your use and analysis of the threshold here relative to prior work? In particular, which part of the threshold analysis fundamentally relies on the welfare objective $v(S) - c(S)$ potentially being negative, rather than following an adaptation of existing budget-feasible valuation-maximization arguments?
>
> **Response:** We first clarify the concern regarding "which part of the threshold analysis fundamentally relies on the welfare objective $v(S) - c(S)$ potentially being negative." In fact, **existing BFM analyses fundamentally rely on the objective being strictly non-negative**. Since our welfare objective lacks this inherent non-negativity, **our analysis must actively circumvent this limitation rather than exploiting "potential negativity."** The potentially negative objective introduces several analytical difficulties, particularly the inability to relax inequalities by dropping non-negative terms (e.g., the commonly used submodular relaxation $f(A)+f(B) \geq f(A\cup B)+f(A\cap B) \geq f(A\cap B)$ may fail, as the last step requires $f(A\cup B) \geq 0$). To overcome this, we introduce the control parameter $\beta$ within the pricing rule, combined with our threshold filtering mechanism, to regulate the value-to-payment ratio. Together, they (1) ensure that the welfare of every candidate solution remains non-negative throughout execution, and (2) establish a controlled relationship between valuation and welfare (i.e., $v(S)\leq g(v(S)-c(S))$ for the candidate solution $S$ and an appropriate $g$), allowing intermediate derivations to leverage the non-negative valuation function and then convert the resulting bounds back into welfare guarantees.
>
> Regarding the novelty of our threshold framework, our design departs from existing BFMs in three major aspects:
>
> 1. **Initialization.** Existing BFMs initialize the threshold at the maximum singleton objective value $\max_u[v(u)]$, which is directly observable for valuation maximization. For welfare maximization, however, the true maximum singleton objective value $\max_u[v(u)-c(u)]$ is unobservable due to private costs, making this approach inapplicable. Naively initializing at the maximum singleton valuation $\max_u[v(u)]$ could set an excessively high threshold, since the welfare of any single element may fall far below this value, leading to arbitrarily poor outcomes. We resolve this by initializing the threshold at an arbitrarily small value $\epsilon$ independent of unknown costs, ensuring no high-welfare element is prematurely rejected.
>
> 2. **Evaluation target.** Existing BFMs (all designed for valuation maximization) evaluate the threshold against the objective value $v(\cdot)$ directly (e.g., checking if $v(S)>\text{threshold}$). For our welfare maximization, the true objective $v(\cdot)-c(\cdot)$ is unobservable, so we cannot filter items by objective value directly. Instead, we evaluate the threshold against the surrogate $v(\cdot)-p(\cdot)$. Since individual rationality enforces $p(u)\geq c(u)$, controlling $v(\cdot)-p(\cdot)$ via the threshold allows us to indirectly guarantee that the selected items contribute high net welfare.
>
> 3. **Growth rate.** Existing clock-auction BFMs universally double the threshold each iteration. Our threshold instead grows by a factor of a carefully optimized $\alpha$, which is mathematically coupled with the value-to-payment ratio parameter $\beta$ and jointly determined by our theoretical analysis to optimize the approximation guarantee (see Theorems 4.8 and 4.10).
>
> We will revise to better clarify these points.

---

> > ### Author Rebuttal · Reviewer_faA1 · 2026-04-04
> >
> > Thank you for your detailed response! I will maintain my positive score.

---

### Decision · Program_Chairs · 2026-04-30

**Decision:**

Accept (regular)

**Comment:**

The paper studies budget-feasible procurement with a submodular valuation function, where the goal is to maximize social welfare rather than the more standard value.  It proposes a constant-factor mechanism with the usual desiderata, answering a natural open question.  All reviewers like the paper, identifying strengths such as significance of the model and the results, novelty in the mechanism and deep technical insights.  There were minor concerns, most of which were resolved after the author response phase.  Overall I believe the paper would be a solid contribution to ICML